



# The PMIP4 contribution to CMIP6 - Part 2: Two Interglacials, Scientific Objective and Experimental Design for Holocene and Last Interglacial Simulations

Bette L. Otto-Bliesner[1], Pascale Braconnot[2], Sandy P. Harrison[3], Daniel J. Lunt[4], Ayako Abe-Ouchi[5,6], Samuel Albani[7], Patrick J. Bartlein[8], Emilie Capron[9,10], Anders E. Carlson[11], Andrea Dutton[12], Hubertus Fischer[13], Heiko Goelzer[14,15], Aline Govin[2], Alan Haywood[16], Fortunat Joos[13], Allegra N. Legrande[17], William H. Lipscomb[18], Gerrit Lohmann[19], Natalie Mahowald[20], Christoph Nehrbass-Ahles[13], Francesco S-R Pausata[21], Jean-Yves Peterschmitt[2], Steven Phipps[22], Hans Renssen[23,24]

[1]National Center for Atmospheric Research, 1850 Table Mesa Drive, Boulder, Colorado 80305, United States of America
[2]Laboratoire des Sciences du Climat et de l'Environnement, LSCE/IPSL, CEA-CNRS-UVSQ, Université Paris-Saclay, F-91191 Gif-sur-Yvette, France
[3]Centre for Past Climate Change and School of Archaeology, Geography and Environmental Science (SAGES), University of Reading, Whiteknights, RG6 6AH, Reading, United Kingdom
[4]School of Geographical Sciences, University of Bristol, Bristol, BS8 1SS, UK.
[5]Atmosphere Ocean Research Institute, University of Tokyo, 5-1-5, Kashiwanoha, Kashiwa-shi, Chiba 277-20 8564, Japan
[6]Japan Agency for Marine-Earth Science and Technology, 3173-25 Showamachi, Kanazawa, Yokohama, Kanagawa, 236-0001, Japan
[7]Institute for Geophysics and Meteorology, University of Cologne, Cologne, Germany
[8]Department of Geography, University of Oregon, Eugene, OR 97403-1251, United States of America
[9]Centre for Ice and Climate, Niels Bohr Institute, University of Copenhagen, Juliane Maries Vej 30, 2100 Copenhagen Ø, Denmark.
[10]British Antarctic Survey, High Cross Madingley Road, Cambridge CB3 0ET, UK
[11]College of Earth, Ocean and Atmospheric Sciences, Oregon State University, Corvallis, OR 97331, United States of America
[12]Department of Geological Sciences, University of Florida, Gainesville, FL 32611
[13]Climate and Environmental Physics, Physics Institute and Oeschger Centre for Climate Change Research, University of Bern, CH-3012 Bern, Switzerland
[14]Institute for Marine and Atmospheric research Utrecht (IMAU), Utrecht University, Princetonplein 5, 3584 CC Utrecht, The Netherlands
[15]Laboratoire de Glaciologie, Université Libre de Bruxelles, CP160/03, Av. F. Roosevelt 50, 1050 Brussels, Belgium
[16]School of Earth and Environment, University of Leeds, Woodhouse Lane, Leeds, West Yorkshire, LS29JT, UK
[17]NASA Goddard Institute for Space Studies, 2880 Broadway, New York, NY 10025, United States of America
[18]Group T-3, Fluid Dynamics and Solid Mechanics, Los Alamos National Laboratory, Los Alamos, NM 87545, United States of America
[19]Alfred Wegener Institute Helmholtz Centre for Polar and Marine Research Bussestr. 24 D-27570 Bremerhaven Germany
[20]Earth and Atmospheric Sciences, Cornell University, Ithaca, NY 14850, United States of America
[21]Department of Meteorology, Stockholm University, 106 91 Stockholm, Sweden
[22]Institute for Marine and Antarctic Studies, Uinversity of Tasmania, Hobart, Tasmania 7001, Australia
[23]Department of Earth Sciences, Vrije Universiteit Amsterdam, De Boelelaan 1085, 1081HV Amsterdam, The Netherlands
[24]Department of Environmental and Health Studies, University College of Southeast Norway, 3800 Bø i Telemark, Norway

*Correspondence to*: Bette L. Otto-Bliesner (ottobli@ucar.edu)





**Abstract.** Two interglacial epochs are included in the suite of Paleoclimate Modeling Intercomparison Project (PMIP4) simulations in the Coupled Model Intercomparison Project (CMIP6). The experimental protocols for Tier 1 simulations of the mid-Holocene (*midHolocene,* 6000 years before present) and the Last Interglacial (*lig127k,* 127,000 years before present) are described here. These equilibrium simulations are designed to examine the impact of changes in orbital forcing at times when atmospheric greenhouse gas levels were similar to those of the preindustrial period and the continental configurations were almost identical to modern. These simulations test our understanding of the interplay between radiative forcing and atmospheric circulation, and the connections among large-scale and regional climate changes giving rise to phenomena such as land-sea contrast and high-latitude amplification in temperature changes, and responses of the monsoons, as compared to today. They also provide an opportunity, through carefully designed additional CMIP6 Tier 2 and Tier 3 sensitivity experiments of PMIP4, to quantify the strength of atmosphere, ocean, cryosphere, and land-surface feedbacks. Sensitivity experiments are proposed to investigate the role of freshwater forcing in triggering abrupt climate changes within interglacial epochs. These feedback experiments naturally lead to a focus on climate evolution during interglacial periods, which will be examined through transient experiments. Analyses of the sensitivity simulations will also focus on interactions between extratropical and tropical circulation, and the relationship between changes in mean climate state and climate variability on annual to multi-decadal timescales. The comparative abundance of paleoenvironmental data and of quantitative climate reconstructions for the Holocene and Last Interglacial make these two epochs ideal candidates for systematic evaluation of model performance, and such comparisons will shed new light on the importance of external feedbacks (e.g., vegetation, dust) and the ability of state-of-the-art models to simulate climate changes realistically.

**Keywords**: paleoclimate simulations, transient climate evolution, climate-system feedbacks, interglacial, model evaluation

# 1 Introduction

The modeling of paleoclimate, using physically based tools, has long been used to understand and explain past environmental and climate changes (Kutzbach and Street-Perrott, 1985), and is increasingly seen as a strong out-of-sample test of the models that are used for the projection of future climate changes (Braconnot et al., 2012; Harrison et al., 2014; Harrison et al., 2015; Schmidt et al., 2014). The Paleoclimate Modelling Intercomparison Project (PMIP) has served to coordinate paleoclimate experiments and data-model comparisons for several decades (Braconnot et al., 2012; Braconnot et al., 2007a; Braconnot et al., 2007b; Joussaume and Taylor, 1995; Joussaume et al., 1999), and now spearheads the paleoclimate contribution to the current phase of the Coupled Model Intercomparison Project (CMIP6, Eyring et al., 2016). Five paleoclimate experiments are included in CMIP6 Tier 1 (Kageyama et al., 2016): two of these experiments focus on comparing the most recent interglacial epochs and specifically the current interglacial (the Holocene) and the previous interglacial (the Last Interglacial, LIG) periods (Fig. 1). These two experiments are of interest because they examine the response of the climate system to relatively simple changes in forcing compared to the present. The main difference in forcing from present was in the



latitudinal and seasonal distribution of incoming solar radiation (insolation) caused by known changes in the Earth's orbit; greenhouse gas (GHG) concentrations were similar to those of the preindustrial period and the continental configurations were also very similar to modern. The changes in insolation are characterized by enhanced seasonal contrast in the northern hemisphere (NH) (and reduced seasonal contrast in the southern hemisphere, SH), giving rise to warmer NH summers and a significant enhancement of the NH monsoons. Differences in orbital configuration between the two interglacial periods (Berger, 1978) mean that these changes are stronger in the LIG

than in the Holocene, but the observational basis for evaluating model simulations is more extensive in the Holocene than the LIG because of preservation issues. Taken together, these two interglacial periods are good test cases of our mechanistic understanding of the interplay between radiative forcing and atmospheric circulation, and opportunities to examine connections among large-scale and regional climate changes which give rise to phenomena such as land-sea contrast and high-latitude amplification of temperature changes, the regulation of atmospheric $CO_2$ and

biogeochemical cycles, and the waxing and waning of the monsoons.

The Tier 1 interglacial experiments for CMIP6 are time-slice (or equilibrium) experiments at 6000 and 127,000 years before present (where present is defined as 1950), hereafter referred to as 6 ka (*midHolocene*) and 127 ka (*lig127k*). The mid-Holocene interval has been the focus for model simulations, model-model comparisons, paleodata synthesis, and model-data comparison since the beginning of PMIP, and this work has contributed to

model evaluation and understanding of climate change in the last three major assessments of the Intergovernmental Panel on Climate Change (Flato et al., 2013; Folland et al., 2001; Hegerl et al., 2007; Jansen et al., 2007; Masson-Delmotte et al., 2013). Systematic benchmarking against pollen-based reconstructions of climate variables and lake-level-based water-balance reconstructions (Braconnot et al., 2012; Braconnot et al., 2007b; Coe and Harrison, 2002; Harrison et al., 2014; Harrison et al., 2015; Harrison et al., 1998) have highlighted that climate models persistently

underestimate changes in the monsoon precipitation and produce too much continental drying (Harrison et al., 2015). This raises questions about the role of systematic model biases on simulated climate changes and on the level of model complexity required to simulate climate changes correctly.

Given the long history of coordinated model experiments for 6 ka, this period allows us to assess whether there is an improvement in the ability of models to reproduce a climate state different from the modern one. For this reason the

Tier 1 *midHolocene* experiment is one of two possible entry cards for PMIP simulations in CMIP6 (Table 1): all modeling groups contributing to PMIP4-CMIP6 must perform either the *midHolocene* experiment or a simulation of the Last Glacial Maximum (Kageyama et al., 2016).

Although the LIG (129 ka to 116 ka) was discussed in the First Assessment Report of the IPCC (Folland et al., 1990), it gained more prominence in the IPCC Fourth and Fifth Assessment (AR4 andAR5) because of

reconstructions highlighting that global mean sea level was at least 5 m higher (but probably no more than 10 m higher) than present for several thousand years (Dutton et al., 2015a; Jansen et al., 2007; Masson-Delmotte et al., 2013). Thus the LIG is recognized as an important period for testing our knowledge of climate-ice sheet interactions in warm climate states. However, the ensemble of LIG simulations examined in the AR5 (Masson-Delmotte et al., 2013) was not wholly consistent: the orbital forcing and GHG concentrations varied between the simulations. While



it has been suggested that differences in regional temperatures between models might reflect differences in cryosphere feedback strength (Yin and Berger, 2012; Otto-Bliesner et al., 2013) or differences in the simulation of the Atlantic Meridional Overturning Circulation (AMOC) (Bakker et al., 2013; Masson-Delmotte et al., 2013), differences between models could also have arisen because of differences in the experimental protocols. Furthermore, the LIG simulations were mostly made with older and/or lower-resolution versions of the models than

were used for future projections, making it more difficult to use the results to assess model reliability (Lunt et al., 2013). The Tier 1 *lig127k* experiment (Table 1) is designed to address the climate responses to stronger orbital forcing than the *midHolocene* experiment using the same state-of-the-art models and following a common experimental protocol. It will provide a basis to address the linkages between ice sheets and climate change in collaboration with the Ice Sheet Model Intercomparison Project for CMIP6 (ISMIP6) (Nowicki et al., 2016).

The *midHolocene* and *lig127k* experiments are starting points for examining interglacial climates. A number of other experiments will be made in the current phase of PMIP (PMIP4) to facilitate diagnosis of these "Tier 1" experiments (Tier 1 experiments are those associated with CMIP6, Tier 2 simulations are sensitivity studies in the framework of PMIP4, but not CMIP6. These will include sensitivity experiments to examine the impact of uncertainties in boundary conditions and the role of feedbacks in modulating the response to orbital forcing. Ocean, vegetation, and

dust feedbacks, and the synergies between them, have been a focus in previous phases of PMIP (Braconnot et al., 1999; Dallmeyer et al., 2010; Otto et al., 2009; Wohlfahrt et al., 2004) and this allows us to design simple experimental protocols to compare the strength of these feedbacks in different climate models. Simulations with prescribed but realistic vegetation cover will be a major focus for both the Holocene and LIG in PMIP4, and comparison of these simulations with ESM simulations that include dynamic vegetation will allow exploration of the

magnitude of land-surface biases in these latter models. Changes in vegetation and land-surface hydrology are an important control on dust emissions (Tegen et al., 2002; Engelstädter et al., 2003), which can affect the strength of the West African Monsoon (Konare et al. 2008, Pausata et al. 2016). The examination of the dust feedback will be a new focus in PMIP4. In addition, the LIG provides an ideal opportunity to examine the role of cryosphere feedbacks through sensitivity experiments, which will be a focus of additional experiments associated with both the Holocene

and the LIG. One such feedback is the release of freshwater into the ocean and the role of such freshwater forcing in generating more abrupt climate changes than would be expected for the smoothly varying changes in insolation forcing during an interglacial (Goelzer et al., 2016a; Luan et al., 2015; Stone et al., 2016). Understanding the role of feedbacks in general on the generation of abrupt climate changes, and the need to understand the relationship between mean climate changes and short-term (annual to multi-decadal) climate variability, leads naturally to a

desire to simulate the transient behavior of the climate system – and such transient experiments will be made for both the Holocene and LIG time periods. New results have highlighted the possibility to use reconstruction of past interannual variability from corals and mullusc shells to assess the Holocene simulated changes in variability at the scale of the tropical Pacific Ocean (Emile-Geay et al. 2016). Groups are also encouraged to run their models with an active land and ocean carbon cycle to assess terrestrial and ocean carbon storage and differences between the two

interglacial periods.





The aim of this paper is to present and explain the experimental design both for the PMIP4-CMIP6 Tier 1 interglacial experiments, and for associated Tier 2 sensitivity and transient experiments. Section 2 describes and discusses the PMIP4-CMIP *midHolocene* entry card and *lig127k* Tier 1 simulations. Section 3 describes Tier 2

PMIP4 sensitivity studies that can be carried out to diagnose these core simulations. Section 4 briefly describes the paleodata resources, which can be used to evaluate the simulations.

## 2 Experimental design for the Tier 1 PMIP4-CMIP6 *midHolocene* and *lig127k* simulations

The core or Tier 1 experiments for the Holocene and the LIG are the *midHolocene* and *lig127k* simulations. The CMIP DECK *piControl* for 1850 C.E and the CMIP6 *historical* experiment  (see Eyring et al. 2016 for description

of these experiments) are the reference simulations to which the paleo-experiments will be compared. Thus, the paleo-experiments must use the same model components and follow the same protocols for implementing external forcings as are used in the *piControl* and *historical* simulations. The *midHolocene* simulation is one of the PMIP entry cards in the PMIP4-CMIP6 experiments, which means that groups who run the *lig127k* simulation must also run either the *midHolocene* or the *lgm* (Last Glacial Maximum) experiment (Kageyama et al., 2016). The boundary

conditions for the *midHolocene*, *lig127k,* and *piControl* experiments are given in Table 1, and more detailed information is given below.

### 2.1 Orbital configuration, solar constant, and insolation anomalies

Earth's orbital parameters (eccentricity, longitude of perihelion, and obliquity) should be prescribed following Berger and Loutre (1991). These parameters affect the seasonal and latitudinal distribution and magnitude of solar

energy received at the top of the atmosphere and, in the case of obliquity, the annual mean insolation at any given latitude (Berger and Loutre, 1991). The DECK *piControl* simulations are to use the orbital parameters appropriate for 1850 C.E (Table 1) (Eyring et al., 2016), when perihelion occurs close to the boreal winter solstice. The exact date slightly varies depending on the internal model calendar and the number of days used to define a year. Because of this and the fact that the length of the seasons varies as a function of precession and eccentricity (Joussaume and

Braconnot, 1997), the vernal equinox must be set to noon on March 21th in all the simulations (*piControl, midHolocene and lig127k*). The orbit at 127 ka was characterized by larger eccentricity than at 1850 C.E., with perihelion occurring close to the boreal summer solstice (Fig. 2). The tilt of the Earth's axis was maximal at 131 ka and remained higher than in 1850 C.E. through 125 ka; obliquity at 127 ka was 24.04° (Table 1). The orbit at 6 ka was characterized by an eccentricity of 0.018682, similar to 1850 C.E. Obliquity was similar though slightly larger

(24.105°) than at 127ka, and perihelion at 6 ka occurred near the boreal autumn equinox. The different orbital configurations for the *midHolocene* and *lig127k* result in different seasonal and latitudinal distribution of top-of-atmosphere insolation compared to the DECK *piControl* (Fig. 3). Both time periods show large positive insolation anomalies during boreal summer. July-August anomalies between 40 and 50°N reach about 55-60 W m$^{-2}$ at 127 ka and 25 W m$^{-2}$ at 6 ka. The higher obliquity at 127 ka and 6 ka contributes to a small but positive annual insolation

anomaly compared to preindustrial at high latitudes in both hemispheres. The global difference in insolation forcing between the interglacial experiments and the preindustrial is negligible.





The difference in orbital configuration between 127 ka, 6 ka and preindustrial means that there are differences in season length that should be accounted for in calculating seasonal changes (Kutzbach and Gallimore, 1988). The bias introduced from using the modern calendar rather than the celestial calendar to calculate seasonal averages is shown in Fig. 2, when the date of the vernal equinox is assigned to March 21 at noon. To be able to account for this effect when comparing the simulations to the paleoclimate reconstructions, daily outputs of at least surface temperature, precipitation and winds must be archived. Programs that provide an approximate estimate of monthly means on the fixed-angular celestial calendar from fixed-day calendar will be available on the PMIP4 web page.

The solar constant prescribed for the *midHolocene* and *lig127k* simulations is the same as in the DECK *piControl* simulation, which is fixed at the mean value for the first two solar cycles of the historical simulation (i.e. 1850-1871) (Eyring et al., 2016). This value (1360.7 W m$^{-2}$) is lower than the value for the solar constant used by some models in PMIP3 (1365 W m$^{-2}$) and this leads to a global reduction of incoming solar radiation compared to the PMIP3 experiments (Fig. 4). The slight differences in orbital parameters between the 1850 CE reference periods to be used for PMIP4-CMIP6 and the 1950 CE reference used for PMIP3 leads to seasonal differences in forcing with a slight decrease in boreal spring and increase in boreal autumn. The combination of the two factors leads to an overall reduction: the largest reduction occurs in boreal spring and is about 1.6 W m$^{-2}$ between 10°S and 40°N.

### 2.2 Greenhouse gases

Ice-core records from Antarctica and Greenland provide measurements of the well-mixed GHGs: $CO_2$, $CH_4$, and $N_2O$ (Fig. 1). These measurements are given as molar mixing ratios in dry air in units of parts per million (ppm) or parts per billion (ppb) respectively. For simplicity, we use the term 'concentration' for these mixing ratios. By 6 ka and 127 ka, the concentrations of atmospheric $CO_2$ and $CH_4$ had increased from their respective levels during the previous glacial periods, the Last Glacial Maximum and the penultimate glaciation, to values comparable to preindustrial levels.

*midHolocene.* In PMIP4-CMIP6, we use a revised version of an earlier trace gas reconstruction (Joos and Spahni, 2008). The $CO_2$ concentration for the mid-Holocene is derived from ice-core measurements from Dome C (Monnin et al., 2001; Monnin et al., 2004) and dated using the AICC2012 age scale (Veres et al., 2013). A smoothing spline (Bruno and Joos, 1997; Enting, 1987) with a nominal cut-off period of 3000 years was used to produce a continuous $CO_2$ record. This yields a $CO_2$ concentration of 264.4 ppm at 6 ka. Methane was measured in ice from Antarctic ice cores EDC (Fluckiger et al., 2002), Dronning Maud Land (EPICA Community Members, 2006) and Talos Dome (TALDICE). For Greenland, methane data are from GRIP (Blunier et al., 1995; Chappellaz et al., 1997; Spahni et al., 2003), GISP2 (Brook, 2009), and GISP2D (Mitchell et al., 2013). Both are splined with a nominal cut-off period of 200 years. This results in a concentration of 574 ppb for the Antarctic ice cores, representative for high latitude Southern Hemisphere air, and of 620 ppb for the Greenland ice cores, representative for the high latitude Northern Hemisphere air, and an estimated global mean value of 597 ppb. The $N_2O$ data around 6 ka are from a compilation of published data from EDC (Fluckiger et al., 2002; Spahni et al., 2005) and new, unpublished data measured at



University of Bern using ice from Greenland (NGRIP) and Talos Dome (TALDICE). The data are splined with a nominal cut-off period of 700 yr and the resulting $N_2O$ concentration at 6 ka is 262 ppb.

The realistic GHG concentrations used for the *midHolocene* PMIP4-CMIP6 experiment are different from those used in the PMIP3 experiments (Braconnot et al. 2012). The PMIP3 experiments were designed simply to examine the effects of changes in orbital forcing, and the $CO_2$ concentrations were therefore kept the same as the value typically used in pre-industrial experiments (280 ppm) although other GHGs were prescribed from ice-core measurements. The use of realistic values for all the GHGs in the PMIP4-CMIP6 *midHolocene* experiment may

improve comparisons with paleoclimate reconstructions and will ensure that the *midHolocene* experiment is consistent with planned transient Holocene simulations (see Section 3). However, the reduction in $CO_2$ concentration from 280 to 264.4 ppm will reduce GHG forcing by about 0.3 W m$^2$ (Myhre et al., 1998), which translates to a difference in global mean surface air temperature of -0.24°C when applying an equilibrium climate sensitivity of 3°C for a nominal doubling of $CO_2$. Simulations with the IPSL model (Dufresne and co-authors, 2013)

show that this change in the experimental protocol between PMIP3 and PMIP4-CMIP6 yields a global mean cooling of 0.24 ± 0.04°C, as expected, but that there are regional differences of up to 0.5°C in parts of Eurasia and in South Africa (Fig. 5). Although these differences are small overall, they will need to be accounted for in comparisons between the PMIP4-CMIP6 *midHolocene* simulations and previous generations of PMIP 6ka simulations.

*lig127k.* The LIG GHG concentrations are available solely from Antarctic ice cores. $CO_2$ concentrations can only be

derived from Antarctic ice, because of potential in-situ $CO_2$ production in the Greenland ice sheet (Tschumi and Stauffer, 2000). We also do not have any reliable $CH_4$ and $N_2O$ concentrations from Greenland in the LIG due to melt layers in the ice, as Greenland temperatures were significantly warmer at that time compared to modern (Fig. 1) (NEEM Community Members 2013). For the *lig127k* simulation (Table 1), we adopt mean values for 127.5-126.5 ka on the AICC2012 age scale (Bazin et al., 2013) from EPICA Dome C (Bereiter et al., 2015; Schneider et al.,

2013) for $CO_2$, from EPICA Dome C and EPICA Dronning Maud Land (Loulergue et al., 2008; Schilt et al., 2010) for $CH_4$, as well as from EPICA Dome C and Talos Dome (Schilt et al., 2010) for $N_2O$. The atmospheric $CO_2$ and $N_2O$ concentrations of 275 ppm and 255 ppb, respectively, can be regarded as globally representative, while the mean ice core $CH_4$ concentration (662 ppb) is representative for high-latitude Southern Hemisphere air. A global mean atmospheric $CH_4$ concentration of 685 ppb is adopted for 127 ka, thereby assuming the same difference (23

ppb) between the global mean atmospheric $CH_4$ and Antarctic ice core $CH_4$ values as for the mid-Holocene.

### 2.3 Paleogeography and ice sheets

*midHolocene.* Several lines of evidence indicate that the ice sheets had their modern characteristics by the mid-Holocene, except in a few places such as the Baffin Islands (Carlson et al., 2008b; Clark et al., 2000). While the presence of a relict of the Laurentide ice sheet may be the origin of model-data mismatches in the climate of eastern

North America (Wohlfahrt et al., 2004), the effect is local and small. Cosmogenic surface exposure ages and threshold lake records (Carlson et al., 2014; Larsen et al., 2015; Sinclair et al., 2016) also suggest that by 6 ka, the Greenland ice sheet was similar in extent to the present. Uncertainties in the reconstructions of this relict ice sheet





would have a larger impact on the simulations, and thus ice sheet distribution and elevations, land-sea mask, continental topography and oceanic bathymetry should all be prescribed as the same as in *piControl* in the
*midHolocene* simulation (Table 1).

*lig127k.* Evidence for the evolution of the ice sheets during the LIG comes mainly from proximal marine records (Carlson and Winsor, 2012). The deposition of a detrital carbonate layer in the Labrador Sea, dated to around 128 ka based on geomagnetic secular variation (Winsor et al., 2012), suggests that ice had retreated from Hudson Bay and is taken to indicate the final demise of the Laurentide ice sheet (Carlson, 2008; Nicholl et al., 2012). The
disappearance of the Eurasian ice sheet is more difficult to constrain because either the proximal marine records lack benthic $\delta^{18}O$ data, or the benthic $\delta^{18}O$ data show trends that are different from those of open ocean records during the LIG (Bauch, 2013). The cessation of deposition of ice-rafted debris (IRD) from the Eurasian ice sheet was dated to between 128-126 ka using $\delta^{18}O$ (Risebrobakken et al., 2006). However, sea-level data (Dutton et al., 2015b) suggests that this ice sheet disappeared earlier and was likely gone by ~127 ka. Proximal marine records of the
Greenland ice sheet document a gradual retreat during the LIG, with minimum extent around 120 ka (Carlson et al., 2008a; Colville et al., 2011; Stoner et al., 1995; Winsor et al., 2012). However, Greenland-sourced IRD reached a minimum similar to the Holocene before ~127 ka (Colville et al., 2011; Winsor et al., 2012).

The extent of the Antarctic ice sheets is not directly constrained by data proximal to the ice sheet at 127 ka. Given higher-than-present sea levels, the gradual retreat of the Greenland ice sheet, and the lack of other NH ice sheets, it
seems likely that the Antarctic ice sheet was smaller than present by ~127 ka (Colville et al., 2011; Dutton et al., 2015a; Dutton et al., 2015b; Mercer, 1978). The existence of ~250 ka Mt. Erebus ash in the ANDRILL site in Ross Sea could indicate a smaller-than-present West Antarctic ice sheet (WAIS) some time after ~250 ka (McKay et al., 2012). The ice-core record from Mount Moulton, West Antarctica could be consistent with deglaciation of much of West Antarctica during the LIG, and likely at 126-130 ka (Steig et al., 2015). Standalone ice sheet model
simulations forced by ocean warming suggest the West Antarctic ice sheet to be a major contributor to LIG global mean sea level rise, with contributions also coming from the marine-based portions of the East Antarctic ice sheet (DeConto and Pollard, 2016). Contributions are 6.0-7.5 m of equivalent sea-level rise, which would explain global mean sea level being at least +6 m by ~127 ka (Dutton et al., 2015b). However, because of the difficulty in implementing ice-to-ocean changes for the WAIS and the uncertainties associated with dating the changes in the
other ice sheets, the paleogeography of the *lig127k* simulation will be prescribed the same as in the DECK *piControl* simulation (Table 1). In view of the greater uncertainty associated with the prescription of ice sheets in the *lig127k* experiment, this aspect of the boundary conditions will be a major focus of sensitivity experiments (see Section 3).

### 2.4 Vegetation

There is abundant evidence for changes in vegetation distribution during the mid-Holocene and the LIG (Goni et al.,
2005; Harrison and Bartlein, 2012; Hely et al., 2014; LIGA Members, 1991; Prentice et al., 2000). However, there is insufficient data coverage for many regions to be able to produce reliable global vegetation maps. Furthermore, given the very different levels of complexity in the treatment of vegetation properties, phenology and dynamics in





the current generation of climate models, paleo-observations do not provide sufficient information to constrain their behavior in a comparable way. The treatment of natural vegetation in the *midHolocene* and *lig127k* simulations
should therefore be the same as in the DECK *piControl* simulation. That is, depending on what is done in the DECK *piControl* simulation, vegetation should either be prescribed to be the same as in that simulation, or prescribed but with interactive phenology, or predicted dynamically (Table 1). Uncertainties related to the treatment of vegetation in the different simulations will be analyzed through sensitivity experiments (see section 3).

### 2.5 Aerosols: tropospheric dust and stratospheric volcanic

Natural aerosols show large variations on glacial-interglacial time scales, with low aerosol loadings during interglacials compared to glacials, and during the peak of the interglacials compared to present day (Albani et al., 2015; deMenocal et al., 2000; Kohfeld and Harrison, 2000). Atmospheric dust affects radiative forcing at a regional scale and can therefore affect precipitation and surface hydrology (Miller et al., 2004; Yoshioka et al., 2007) including the monsoons (Konare et al., 2008; Pausata et al., 2016; Vinoj et al., 2014) as well as moderating snow
albedo feedbacks when sufficient dust is deposited (Krinner et al., 2006). While model simulations that are observationally constrained by a global compilation of dust records suggest that the global dust budget was dominated by NH dynamics during the *midHolocene* as it is today, the regional patterns of dust loading were different (Albani et al., 2015). This motivates the inclusion of changes in dust loading in the *midHolocene* and *lig127k* simulations (Table 1, Figure 6).

As in the case of vegetation, the implementation of changes in atmospheric aerosol in the *midHolocene* and *lig127k* simulations should follow the treatment used for the DECK *pre-industrial* and *historical* simulations. Models with an interactive representation of dust should prescribe changes in soil erodibility or dust emissions to account for the changes in dust sources during the interglacials (datasets available at https://pmip4.lsce.ipsl.fr/doku.php/exp_design:mh). Although the maps provided by PMIP for this purpose are for
mid-Holocene conditions and from the only model simulation available (Albani et al., 2015) it should be used for both the *midHolocene* and *lig127k* simulations. For each model configuration, if atmospheric dust loading is prescribed in the DECK *piControl* and *historical* simulations, the *midHolocene* and *lig127k* simulations should use the three-dimensional monthly climatology of atmospheric dust mass concentrations or aerosol optical depths available from the same data-constrained simulations as the soil erodibility maps. Also available are datasets of the
dust shortwave and longwave direct radiative forcing. If atmospheric dust loading is not represented in the DECK *piControl* and *historical* simulations, it should not be included in the *midHolocene* and *lig127k* simulations. The impact of dust on the radiation balance is sensitive to the optical properties prescribed (Perlwitz et al., 2001); it is uncertain how optical properties might change during interglacials (Potenza et al., 2016; Royer et al., 1983). Uncertainties in the protocol and in the interplay between dust and vegetation will be a focus of the analyses.

There is no observationally-constrained estimate of the volcanic stratospheric aerosol for either the mid-Holocene or the LIG. The background volcanic stratospheric aerosol used in the CMIP6 DECK *piControl* should be used for the



*midHolocene* and *lig127k* simulations. Other aerosols included in the DECK *piControl* should similarly be included in the *midHolocene* and *lig127k* simulations.

### 2.6 Setup and documentation of simulations

Spin-up procedures differ for different models, but the model must be run for long enough to avoid long-term drift in the global energetics and major climate variables, including intermediate ocean temperatures. A minimum of 500 years for the total length of simulation is required, assuming an initialization from modern ocean conditions. The outputs stored in the CMIP6 database should be representative of the equilibrium climates of the *midHolocene* and *lig127k* time periods. A minimum of 100 years of output is required to be uploaded for each simulation (usually the

final 100 years of the simulation). However, given the increasing interest in analyzing multi-decadal variability (e.g. Wittenberg, 2009) and the availability of reconstructions of ENSO (El Niño-Southern Oscillation) and other modes of variability (see Sect. 3), modeling groups are encouraged to provide outputs for at least 500 years if possible.

The simulations should follow the CMIP6 data request and format. For groups only contributing to PMIP, the data

format and organization on the ESGF archive is the same as for CMIP6, except that the provision of daily values can be limited to 2D surface variables, including temperature, precipitation and winds. Groups are also asked to keep a 20 year period with all the output needed to force regional area-limited models, since we would like to strengthen the linkages between global and regional simulations for regional model-data comparisons.

The required detailed documentation of the PMIP4-CMIP6 simulations is documented in Kageyama et al., 2016.

Documentation should be provided via the ESDOC website and tools provided by CMIP6 (http://es-doc.org/) to facilitate communication with other CMIP6 projects. This documentation should be mirrored on the PMIP4 website to facilitate linkages with non-CMIP6 simulations to be carried out in PMIP4. For the *midHolocene* and *lig127k*, the documentation should include:

- A description of the model and its components;

- Information on the implementation of the forcings. The provision of figures and tables giving monthly-latitude insolation anomalies and daily incoming solar radiation at the top of the atmosphere (TOA) for one year should be provided because this allows the implementation of the most critical forcing to be checked. Information about the implementation of aerosols should also be provided. Any differences from the protocols in Table 1 need to be documented;

- Information about the initial conditions and spin-up technique used, as well as any information about model tuning that could affect albedo, climate thresholds or climate variability. A measure of the changes and drifts in key variables (e.g., globally averaged 2m temperatures, sea-surface temperatures, bottom ocean temperatures, and top-of-the-atmosphere radiative fluxes) should be provided.

### 3 PMIP4-CMIP6 Tier 2 and Tier 3 Simulations



The selection of only two intervals, *midHolocene* and *lig127k*, for PMIP4-CMIP6 interglacial experiments is designed to maximize both the model ensemble size and opportunities for model evaluation, since both periods have been the focus for data synthesis. However, this means that the experiments do not sample the diversity in the transient forcings and responses during the LIG and the Holocene. Although transient simulations for these two periods are included in the suite of PMIP4 simulations (see 3.5), there is utility in examining other interglacial

climates using equilibrium experiments parallel to the *midHolocene* and *lig127k* simulations, particularly in order to provide additional samples of the response of the system to insolation forcing. Additional, Tier 2 experiments – the end of the LIG (116 ka) and the early Holocene (9.5 ka) (see 3.1) – are proposed to address this.

        Uncertainties in the boundary and initial conditions for the mid-Holocene and LIG mean that the PMIP4-CMIP6

*midHolocene* and *lig127k* simulations may not capture important feedbacks accurately. The major sources of uncertainty in the boundary conditions are the prescription of modern vegetation cover by some models, and the prescription of modern ice sheets in the *lig127k* simulation. Both sources of uncertainty can be addressed through Tier 2 sensitivity experiments (see 3.2, 3.3). The equilibrium experiments also do not address climate changes forced by the non-linear behavior of ice sheet-ocean coupling, or the possibility that such feedbacks could give rise

to abrupt changes in climate superimposed on the more slowly-varying insolation forcing during the Holocene and the LIG. This will be addressed through Tier 2 idealized simulations of specific freshwater-forcing events, specifically the Heinrich 11 event at the beginning of the LIG and the 8.2 ka event during the Holocene (see 3.4). Other feedbacks could give rise to more abrupt responses to orbital forcing. Understanding the interplay among different components of the Earth system in determining the long-term evolution of LIG and Holocene climate is the

major goal of the proposed Tier 3 transient experiments (Section 3.5) to be carried out during PMIP4.

        Further information and access to datasets is available on PMIP4 web site and will be updated during the course of the project (https://pmip4.lsce.ipsl.fr/doku.php/exp_design:index)

### 3.1 Equilibrium response to alternative states of orbital forcing

***Early Holocene***. The maximum expression of orbitally-induced differences in TOA insolation forcing from present occurred during the early part of the Holocene, but the climate at this time was still affected by the presence of a relict of the Laurentide ice sheet (Carlson et al., 2008b). As a result, summer temperatures in mid- to high latitudes were cooler than during the mid-Holocene (Carlson et al., 2008b; Renssen et al., 2012; Renssen et al., 2009). The presence of the ice sheet delayed the response to insolation forcing in monsoon regions (Lezine et al., 2011; Marzin

et al., 2013). It has also been suggested that the remnant ice-sheet may have counteracted the reduction of ENSO variability in response to orbital forcing in the early Holocene (Carre et al., 2014; Luan et al., 2015). Protocols for early Holocene experiments were developed in previous phases of PMIP (PMIP2, PMIP3), and provide the basis for proposed PMIP4 simulation for 9.5 ka. Since the phase of precession at 9.5 ka is similar to that of 127k, this experiment provides a basis for examination of the similarities in seasonal changes between the two interglacials

(Braconnot et al., 2008). Following the experimental protocol for the *midHolocene* simulation, orbital parameters



should be changed following Berger and Loutre (1991). The extent and topography of the ice sheet should be prescribed using either ICE-6G or GLAC-1D, as proposed by the PMIP deglaciation working group (Ivanovic et al., 2016). GHG concentrations can also be prescribed from the last deglaciation experiment.

***Lig116k*** Continental ice sheet growth and associated sea level lowering started at ~116 ka, marking the end of the LIG (Stirling et al., 1998). Simulations with climate models that include feedbacks among the atmosphere, ocean, land, and sea ice are able to simulate sufficient cooling to initiate ice sheet growth when forced with the 116 ka orbital conditions reducing NH summer insolation (Herrington and Poulsen, 2012; Jochum et al., 2012). However, the result is sensitive to the atmospheric $CO_2$ concentration used. In order to test the sensitivity to $CO_2$, we

propose sensitivity experiments using orbital parameters for 116 ka (*lig116k*). In the first experiment, the $CO_2$ concentration should be prescribed as 280 ppm, while in the second experiment it should be set to 240 ppm. All other forcings and boundary conditions will remain the same as the *lig127k* simulation.

### 3.2 Sensitivity to Prescribed Vegetation

Except in the case of models with dynamic vegetation, the *midHolocene* and *lig127k* simulations will be run with

prescribed preindustrial vegetation cover because of the lack of a comprehensive and reliable global data set of vegetation for the two periods. However, pollen and macro-fossil evidence show that boreal forest extended farther north than today in the mid-Holocene (Bigelow and al., 2003; Prentice et al., 2000) and, except in Alaska and central Canada, extended to the Arctic coast during the LIG (Edwards et al., 2003; LIGA, 1991; Lozhkin and Anderson, 1995). Pollen and other biogeographical/geomorphological evidence also indicate northward extension of

vegetation into modern-day desert areas, particularly in northern Africa, both in the mid-Holocene (Drake et al., 2011; Hely et al., 2014; Prentice et al., 2000) and during the maximum phase of the LIG (Castaneda et al., 2009; Hooghiemstra et al., 1992). Given the impact of increased woody cover on albedo and evapotranspiration, these vegetation changes should have profound impacts on the surface energy and water budgets and may help to explain mismatches between simulated and reconstructed high-latitude (Muschitiello et al., 2015) and monsoon climates

(Braconnot et al., 1999; Claussen and Gayler, 1997; Pausata et al., 2016) in both time periods.

We propose a series of sensitivity experiments to explore the feedbacks between vegetation and climate in which vegetation cover in the high-latitudes is changed from tundra to boreal forest (experiment a) and the Sahara desert is (Lozhkin and Anderson, 1995) replaced by evergreen shrub to 25°N and savanna/steppe poleward of 25°N

(experiment b). Ideally, these regional changes should be made separately in order to diagnose the interaction between high-latitude and low-latitude climates, and a third experiment could be made implementing both changes (experiment c). A more realistic simulation of the influence of mid-Holocene vegetation changes in the Sahara (experiment d) could be made using the gridded data set of land-surface conditions provided by Hoelzmann et al. (Hoelzmann et al., 1998), which gives proportions of specific vegetation types (steppe, savanna, xerophytic

woods/scrub, tropical deciduous forest, and tropical montane evergreen forest), open water (lakes), and wetlands on a 1° grid. This data set could also be used for a LIG sensitivity experiment, on the assumption that this provides a



minimum estimate of the changes during that time (see e.g. Drake et al., 2011). The maps provided by Bigelow et al. (2003) provide more nuanced vegetation changes in high-latitude vegetation (experiment e), since they distinguish cold deciduous boreal forests from evergreen boreal forests and tundra. In each experiment, all other boundary

conditions should be implemented as in the baseline *midHolocene* and *lig127k* simulations.

Several modelling groups will be running simulations with models including dynamic vegetation, and this makes it possible to test the impact of the mid-Holocene vegetation changes on modern climate. For groups using dynamic vegetation, an additional sensitivity test is proposed in which the *piControl* simulation is run with the dynamic

vegetation component is switched off and vegetation is prescribed using the Hoelzmann et al. vegetation map. Sensitivity experiments will also be required to characterize the uncertainties related to the prescription of dust fields in the midHolocene and LIG simulations, but it is difficult to anticipate the form of such experiments until the Tier 1 experiments are diagnosed. A first step could be to investigate the vegetation feedback on emission in simulations with interactive dust exploiting the vegetation sensitivity analyses.

**3.3 Sensitivity to Prescribed Ice Sheets**

The *midHolocene* and *lig127k* simulations will be run with prescribed modern ice sheets and paleogeography. However, it is highly likely that the Antarctic ice sheet was smaller than present by ~127 ka, most probably because of the disappearance of the WAIS, and that the Greenland ice sheet was reduced in extent compared to present. Given that only about 3-4 m sea level rise are covered by contributions from ocean thermal expansion (McKay et al.,

2011), land based glaciers (Marzeion et al., 2012), and melting of the Greenland Ice Sheet (Dahl-Jensen et al., 2013; Masson-Delmotte et al., 2013), the remaining sea level rise is most likely to be linked to a mass loss from the Antarctic Ice Sheet. We propose a sensitivity experiment to test the impact of a smaller-than-present Antarctic ice sheet, using a reduced ice-sheet configuration obtained from off-line simulations with their own models or the model results such as those from DeConto and Pollard (2016) or Sutter et al. (2016). These authors used a dynamic ice

sheet model, forced with climate model output and calibrated to reproduce LIG sea-level estimates, to simulate the Antarctic ice sheet at 128 ka. All other boundary conditions should be implemented as in the baseline *lig127k* simulation. Additional simulations in which the Greenland ice sheet is configured to its minimum LIG extent are also of interest, using configurations obtained from off-line simulations, for example from ISMIP6.

**3.4 Freshwater Forcing**

***Sensitivity to the H11 meltwater event during the early LIG.*** Heinrich layers in the North Atlantic, containing high concentrations of IRD, record multiple examples of prolonged iceberg discharge during the past 500 ka (Hemming, 2004; Marino et al., 2015; McManus et al., 1999). Heinrich event 11 (H11) is a well-documented example that occurred from ~135-128 ka (Marino et al., 2015). The associated freshwater flux has been estimated as peaking at ~0.3 Sv at ~132.5 ka and tapering off thereafter (Marino et al., 2015), and is broadly consistent with an estimate of

0.19 Sv at 130 ka based on coral records (Carlson, 2008). There is also evidence of a rapid sea level rise associated with this meltwater pulse, estimated at ~70 m or 28±8 m ka$^{-1}$ during the deglacial transition (Grant et al., 2012).





Model simulations have shown that the freshwater forcing of H11, including its cessation, may be important for explaining the evolution of climate through the early part of the LIG (Goelzer et al., 2016b; Holden et al., 2010; Loutre et al., 2014; Stone et al., 2016). We propose a sensitivity experiment to examine the impact of the H11 event.

The insolation anomalies at 130 ka are similar to those at 127 ka. Therefore the experiment can be made by adding a persistent flux of 0.2 Sv to the North Atlantic between 50 and 70°N for 1000 years, with all other boundary conditions implemented as in the baseline *lig127k* simulation.

***Sensitivity to the 8.2 ka fresh water event during the early Holocene.*** While the climate impact of the 8.2 ka event

is well documented, the magnitude of the freshwater forcing and its duration are less well constrained. There are generally thought to be two components to the freshwater forcing in the early Holocene, a background flux from the Laurentide ice sheet (Hillaire-Marcel et al., 2007; Licciardi et al., 1999) and catastrophic flux from the drainage of Lake Agassiz (Barber et al., 1999; Clarke et al., 2004; Teller et al., 2002). The background flux is small (ca 0.13 Sv) but persistent for several hundred years (Carlson et al., 2009; Carlson et al., 2008b; Clarke et al., 2009; Hillaire-

Marcel et al., 2007). Lake Agassiz appears to have drained in several flood events of relatively short duration, but with an estimated total discharge into the Labrador Sea of ca 151,400 km$^3$ (Andrews et al., 1999; Andrews et al., 1995; Clarke et al., 2009; Clarke et al., 2004; Ellison et al., 2006; Hillaire-Marcel et al., 2007; Kerwin, 1996; Lajeunesse and St-Onge, 2008; Lewis et al., 2012; Roy et al., 2011). The proposed sensitivity experiment uses the orbital, ice sheet, and GHG boundary conditions of an 8.5 ka experiment and imposes a background freshwater flux

of 0.05 Sv for 500 years and a single input of 2.5 Sv for one year. This freshwater flux is added to the Labrador Sea, but modeling groups can chose whether to add it uniformly over the whole of the Labrador Sea or only over part of the area. The simulation is then run until there is evidence for the recovery of the Atlantic Meridional Overturning Circulation (AMOC).

### 3.5 Transient Holocene and LIG simulations

Transient simulations provide an opportunity to examine the time-dependent evolution of climate in response to forcings and feedbacks. Transient simulations of the last deglaciation are a major focus in PMIP4 (Ivanovic et al., 2016). These simulations will be run for the period 21 to 9 ka with time-varying orbital forcing, greenhouse gases, ice sheets and other geographical changes. The later part of this experiment is obviously of interest for comparison with the early Holocene experiments. However, we are also proposing transient simulations focusing on the

Holocene and the LIG.

Using the PMIP-CMIP6 *midHolocene* simulation as a starting point, we propose a transient simulation of the last 6000 years. In this simulation, both orbital parameters and GHGs will be changed following Berger and Loutre (1991) and ice-core measurements (as described in Section 2.2). Changes in paleotopography over the past 6 ka are small and, for simplicity and consistency with the *midHolocene* simulation, we propose using modern values

throughout. Vegetation and aerosols will also be fixed at preindustrial values, except for groups running fully dynamic vegetation and/or aerosol models where the initial state of these components will be derived from their *midHolocene* simulation. Alternatively, some groups may start the Holocene transient simulation from the end of the





last deglaciation experiment at 9 ka, incorporating changes in the evolution of ice sheets and paleotopography

consistent with that experiment. A proposed LIG transient simulation will be run from 128 to 122 ka, using

appropriate changes in orbital forcing but with all other boundary conditions specified as in the *lig127k* simulation.

These simulations as well as simulations planned by some modeling groups with climate-ice sheet models will be

important as input for addressing the role of coupling between climate and the ice sheets.

**4 Paleoenvironmental data and climate reconstructions for comparison to model simulations**

The ability to evaluate the realism of the core PMIP4-CMIP6 simulations and the various sensitivity experiments is

central to PMIP. Some paleoenvironmental observations can be used for direct comparison with model outputs,

including e.g. simulated water balance against lake-level reconstructions (e.g., Coe and Harrison, 2002) or simulated

vegetation patterns against pollen-based vegetation reconstructions (e.g., Perez Sanz et al., 2014). Such qualitative

comparisons are often adequate to evaluate simulations when, as is the case with regional climate changes in the

mid-Holocene and LIG, the changes are large and regionally coherent (Harrison and Bartlein, 2012). There are also

quantitative reconstructions of climate variables from a wide variety of archives. There are uncertainties associated

with such reconstructions (Harrison et al., 2016), both statistical and resulting from an incomplete understanding of

the climate controls on specific types of records, and these uncertainties need to be taken into account in

comparisons with simulations. However, an increasing number of process-based models can be used to translate

climate model outputs into explicit simulations of specific paleo-records (Emile-Geay and Tingley, 2016; Li et al.,

2014; Thompson et al., 2011), allowing uncertainties in process understanding to be made explicit. Drawing on

ongoing work for the LGM and the use of ocean biochemistry, tracer and isotopic modeling, efforts will be made to

isolate key features of the ocean reconstructions that should be reproduced by climate models.

The major analytical focus for the Holocene experiments is on systematic benchmarking (Harrison et al., 2015) of

the core *midHolocene* simulation, analysis of feedbacks, and elucidation of the relationship between mean climate

state and interannual to centennial variability. Analysis of the *midHolocene* simulation and associated sensitivity

experiments benefits from the fact that there has been a major focus on data synthesis for this time period (Bartlein

et al., 2011; Bigelow and al., 2003; Daniau et al., 2012; Emile-Geay et al., 2016; Hessler et al., 2014; Kohfeld and

Harrison, 2000; Leduc et al., 2010; Marchant et al., 2009; Marlon et al., 2013; Pickett et al., 2004; Prentice et al.,

2000). Thus the number of records and spatial coverage of quantitative reconstructions are relatively extensive

(Bartlein et al., 2011; Hessler et al., 2014). There are gaps in coverage from continental regions, particularly in the

SH, but this situation is likely to improve in the near future (Flantua et al., 2015; Herbert and Harrison, 2016).

Knowledge of ocean conditions during the mid-Holocene is poor and likely to remain so, in part because of

incomplete understanding of the causes of differences between sea-surface temperature reconstructions based on

different biological groups and in part because the signal-to-noise ratio in the reconstructions is large due to other

methodological uncertainties (Hessler et al., 2014; Jonkers and Kucera, 2015; Rosell-Mele and Prahl, 2013).  There

are several sources of information about short-term climate variability during the Holocene, including tree-ring

records, spelothems, corals and molluscs. However, there are major gaps in data coverage from the tropical oceans



that challenge our understanding of ENSO variability; the distribution of speleothem records is limited to karst
areas; and few tropical trees show clear-cut seasonality in growth. More comprehensive syntheses of these data are
needed, and there are major challenges in combining the different data sources to yield large-scale reconstructions of
climate variability. It will also be necessary to develop appropriate methods to use these data for comparison with
simulations, focusing on temporal statistics and teleconnection patterns (Emile-Geay et al., 2016; Emile-Geay and
Tingley, 2016).


There are many individual records documenting the evolution of climate through the Holocene, including
quantitative climate reconstructions (Wanner et al., 2008). Synthetic products have either focused on reconstructions
of global temperature changes (Clark et al., 2012; Marcott et al., 2013; Shakun et al., 2012), or are available as
geographically explicit data sets only for a limited number of climate variables in a few regions such as North
America or Europe (Davis et al., 2003; Gajewski, 2015; Mauri et al., 2014; Viau and Gajewski, 2009; Viau et al.,
2006). The only exception to this is the Global Lake Status Data Base (Kohfeld and Harrison, 2000), which provides
qualitative estimates of the change in lake water balance through time globally. The reliability of global temperature
estimates depends on the representativeness of the data included; this point has been made abundantly clear from
comparisons of records of hemispheric temperature changes during the last millennium (Fernandez-Donado et al.,
2013; Moberg, 2013). Currently available reconstructions of global temperature changes during the Holocene are
heavily biased towards marine records, making it imperative that the reliability of these records is assessed using
continental reconstructions (Davis et al., 2015; Liu et al., 2014). The lack of geographically explicit reconstructions
for tropical regions and the SH would limit analysis of the Holocene transient simulations, but efforts are underway
to improve this situation.


The LIG is the most suitable of the pre-Holocene interglacial periods as a focus in PMIP4-CMIP6 because of the
relative wealth of data compared to earlier interglacial periods. However, there is an order of magnitude less
information than for the Holocene, and there are larger uncertainties in dating of specific events. This means that the
LIG data-model comparisons will focus on large-scale features, such as the strength of the high-latitude
amplification of warming and the role of snow and sea-ice feedbacks in this warming. There will also be a major
focus on the tropical water cycle. These analysis will exploit available datasets for the LIG which mostly document
surface sea and air temperatures across the globe (Anderson et al., 2006; Bakker et al., 2013; Brewer et al., 2008;
Capron et al., 2014; McKay et al., 2011; Turney and Jones, 2010) although recent efforts also synthesize
reconstructions of sea ice changes (Esper and Gersonde, 2014; Sime et al., 2013), of the deep ocean circulation
(Oliver et al., 2010), and to a lesser extent the tropical hydrological cycle (Govin et al., 2014). In addition, several
existing maps are reporting vegetation changes in the NH high latitudes (Bennike et al., 2001) and changes in lake
area in the Sahara (Petit-Maire, 1999).

There are also syntheses of quantitative climate reconstructions for the LIG (Turney and Jones, 2008; McKay et al.,
2011), which have been used for model evaluation (Lunt et al., 2013; Otto-Bliesner et al., 2013). The major



limitation in using these two data sets for analysis of the *lig127k* simulations and associated sensitivity experiments is that they are compilations of information about the maximum warmth during the LIG. Given that warming was not synchronous globally (Bauch and Erlenkeuser, 2008; Cortese et al., 2007; Dahl-Jensen et al., 2013; Govin et al., 2012; Masson-Delmotte et al., 2010; Mor et al., 2012; Winsor et al., 2012), these syntheses do not represent a

specific time slice. A more recent compilation by Capron et al. (2014) has used harmonized chronologies for ice and marine records to produce records of the change in high-latitude temperature compared to present for four 2000-year long time slabs, and this approach has been expanded to include the fifth time slab (126-128 ka) for comparison with the *lig127k* simulation (Capron et al., 2016). However, even though these compilations are based on harmonized chronologies, dating uncertainties during the LIG can still be several thousand years depending on the

type of archive and the dating methods (Govin et al., 2015). Furthermore, the different response scales of different components of the climate system means that records from the 126-128 ka time slab may still bear the imprint of the previous deglaciation (Fig. 1) (Capron et al., 2016). In any case, and as with the early Holocene experiments, the *lig127k* simulation will not solely reflect the insolation forcing. It is therefore recommended that data-model comparisons focus on using the temporal evolution of climate, as captured in the Capron et al. (2014) data sets, to

assess the plausibility of the *lig127k* simulation.

## 5 Conclusions

The PMIP4-CMIP6 *midHolocene* and *lig127k* simulations provide an opportunity to examine the impact of two different changes in radiative forcing on climate at times when other forcings were relatively similar to present. Together with planned sensitivity experiments, this focus on the two interglacials will allow us to explore the role of

feedbacks in the climate system and to quantify their contribution to large-scale phenomena relevant to future projections such as land-sea contrast and high-latitude amplification of temperature changes. They will also allow us to address the implications of changes in forcing and feedbacks for the tropical circulation and monsoons – again an issue that is relevant to interpreting future projections. Given that both periods have been foci for model-model and data-model comparisons during previous phases of PMIP, a major focus during CMIP6 will be on evaluating the

realism of the *midHolocene* and *lig127k* simulations using a wide range of paleoenvironmental data and paleoclimate reconstructions. This evaluation will be a direct out-of-sample test of the reliability of state-of-the-art models to simulate climate changes, and particularly the climate warming.

PMIP4 will collaborate with other CMIP6 projects (Kagayema et al., 2016, Table 3). The output from the *lig127k* simulation, for example, will be used to force standalone ice sheet experiments (*ism-lig127k-std*) in ISMIP6. This

will complement the suite of standalone ISMIP6 ice sheet experiments (Nowicki et al., 2016; http://www.climate-cryosphere.org/activities/targeted/ismip6) for the recent past and future and will add to increase our understanding of the ice-sheet sensitivity to climate changes. The PMIP4-CMIP6 *midHolocene* and *lig127k* simulations, and associated sensitivity experiments, are also relevant to analyses of sea-ice feedbacks to climate in SIMIP (Notz et al., 2016) and to assessments of the role of dust forcing by AerChemMIP (Collins et al., 2016). Beyond CMIP6, the

planned PMIP4-CMIP6 interglacial simulations are relevant to the Grand Challenges set by the World Climate Research Programme (WCRP). Both the *midHolocene* and the *lig127k* simulations are relevant to the Grand





Challenge "Clouds, Circulation and Climate Sensitivity", which has a major focus on the controls on the monsoon circulation. Also, the *lig127k* simulation is particularly relevant to the Grand Challenge "Melting Ice and Global Consequences", which addresses the stability of the ice sheets. Those simulations carried out with biogeochemical
cycles enabled are relevant to the Grand Challenge "Carbon Feedbacks in the Climate System".

## 6  Data availability

The forcing and boundary condition data sets described in this paper are available in the PMIP4 repository https://pmip4.lsce.ipsl.fr/doku.php/exp_design:index

**Acknowledgements.** BLO-B acknowledges the funding by the U.S. National Science Foundation of the National
Center for Atmospheric Research. PB, SPH, and FP acknowledge funding from the JPI-Belmont project "PAleao-Constraints on Monsoon Evolution and Dynamics (PACMEDY)" through their national funding agencies. SPH also acknowledges funding from the European Research Council for "GC2.0: Unlocking the past for a clearer future". FJ and CNA acknowledge support by the Swiss National Science Foundation. AH acknowledges funding support from the European Research Council under the Netherlands Earth System Science Centre (NESSC). EC is funded by the
Dutch Ministry of Education, Culture European European Union's Seventh Framework Programme for research and Science (OCW) innovation under Grant nr. 024.002.001the Marie Skłodowska-Curie(FP7/2007–2013)/ERC grant agreement no 600207. This work is a contribution to the PAGES/PMIP QUIGS Working Group.





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





**Table 1. Forcings and boundary conditions. More details can be found in the Section numbers indicated in parentheses.**

|  | 1850 C.E. (DECK *piControl*)[1] | 6ka (*midHolocene*)[2] | 127ka (*lig127k*)[2] |
|---|---|---|---|
| Orbital parameters (2.1) | CMIP DECK *piControl* |  |  |
| Eccentricity | 0.0167643 | 0.018682 | 0.039378 |
| Obliquity (degrees) | 23.459277 | 24.105 | 24.040 |
| Perihelion - 180 | 100.32687 | 0.87 | 275.41 |
| Vernal equinox | Fixed to noon on March 21 | Fixed to noon on March 21 | Fixed to noon on March 21 |
| Greenhouse gases (2.2) |  |  |  |
| Carbon dioxide (ppm) | 284.6 | 264.4 | 275 |
| Methane (ppb) | 808 | 597 | 685 |
| Nitrous oxide (ppb) | 273 | 262 | 255 |
| Other GHG gases | CMIP DECK *piControl* | 0 | 0 |
| Solar constant (Wm$^{-2}$) (2.1) CMIP DECK *piControl* | TSI: 1360.747 SSI, ap if needed | Same as *piControl* | Same as *piControl* |
| Paleogeography (2.3) | Modern | Same as *piControl* | Same as *piControl* |
| Ice sheets (2.3) | Modern | Same as *piControl* | Same as *piControl* |
| Vegetation (2.5) | CMIP DECK *piControl* | Prescribed or interactive as in *piControl* | Prescribed or interactive as in *piControl* |
| Aerosols (2.6) Dust, Volcanic, etc. | CMIP DECK *piControl* | Prescribed or interactive as in *piControl* | Prescribed or interactive as in *piControl* |

[1] More information on the CMIP DECK *piControl* and CMIP6 *historical* protocols can be found at:

http://www.wcrp-climate.org/wgcm-cmip/wgcm-cmip6

[2] Datasets for *midHolocene* and *lig127k* are available on the PMIP4 web page:





**Table 2. Summary of PMIP4 Tier 2 sensitivity simulations complementing PMIP4/CMIP6 Tier 1 interglacial experiments.** More details can be found in the Section numbers indicated in parentheses.

| PMIP4-CMIP6 Tier 1 simulations | | |
|---|---|---|
| | Entry card: *midHolocene* | *lig127k* |
| **PMIP4-CMIP6 sensitivity experiments: Tier 2 simulations** | | |
| Experiments | Holocene | Last Interglacial |
| Orbital forcing and trace gases (3.1) | *hol9.5k*: Early Holocene <br>• Orbital: 9.5 ka<br>• Ice sheet: ICE-6G or GLAC-1D reconstruction[1]<br>• GHG: same as for the deglaciation experiment[1] | *lig116k*: Glacial inception<br>• Orbital: 116 ka<br>• $CO_2$: 280, 240 ppm<br>• Other forcings and boundary conditions: as for *lig127k* |
| Sensitivity to vegetation (3.2) | *midHolocene_veg* <br><br>• prescribed boreal forests in Arctic and shrub/savanna over Sahara<br>• vegetation reconstructions[2]<br>• *midHolocene* equilibrium veg with dgvm in *piControl* | *lig127k_veg* <br><br>• prescribed boreal forests in Arctic and shrub/savanna over Sahara |
| Sensitivity to Ice-Sheet (3.3) | | *lig127k_ais* and *lig127k_gris*<br>• Antarctic ice sheet at its minimum LIG extent<br>• Greenland ice sheet at its minimum LIG extent |
| Test to freshwater flux (3.4) | *hol8.2k:* 8.2 ka event <br><br>• Orbital: 8.2 ka<br>• Ice sheet: ICE-6G or GLAC-1D reconstruction[1]<br>• GHG: same as for the deglaciation experiment[1]<br>• Initial state: 8.5 ka simulation<br>• Meltwater flux of 2.5 Sv for one year added to the Labrador Sea plus 0.05 Sv for 500 years<br>• Run length: preferably until evidence for the recovery of the AMOC. | *lig127k_H11*: Heinrich 11 meltwater event<br>• Meltwater flux of 0.2 Sv to the North Atlantic between 50 and 70°N for 1000 years<br>• Other forcings and boundary conditions: as for *lig127k*<br>• Initial state: *lig127k* simulation |
| **PMIP4-CMIP6 sensitivity experiments: Tier 3 simulations** | | |
| Transient simulations (3.5)<br><br>(Note : Exploratory and flexible set up) | *past6k*: transient Holocene<br>• Transient evolution in Earth's orbit and trace gases<br>• Other boundary conditions (land use, solar, volcanism) may be considered by some groups<br>• Initial state: *midHolocene* | *lig128to122k*: transient LIG<br>• Transient evolution in Earth's orbit and trace gases<br>• Other boundary conditions (ice sheets) may be considered by some groups<br>• Initial state: *last127k* |

[1]Ivanovic et al., 2016; available on the PMIP4 web page
       [2]Hoelzmann et al., 1998; Bigelow et al., 2003; available on the PMIP4 web page



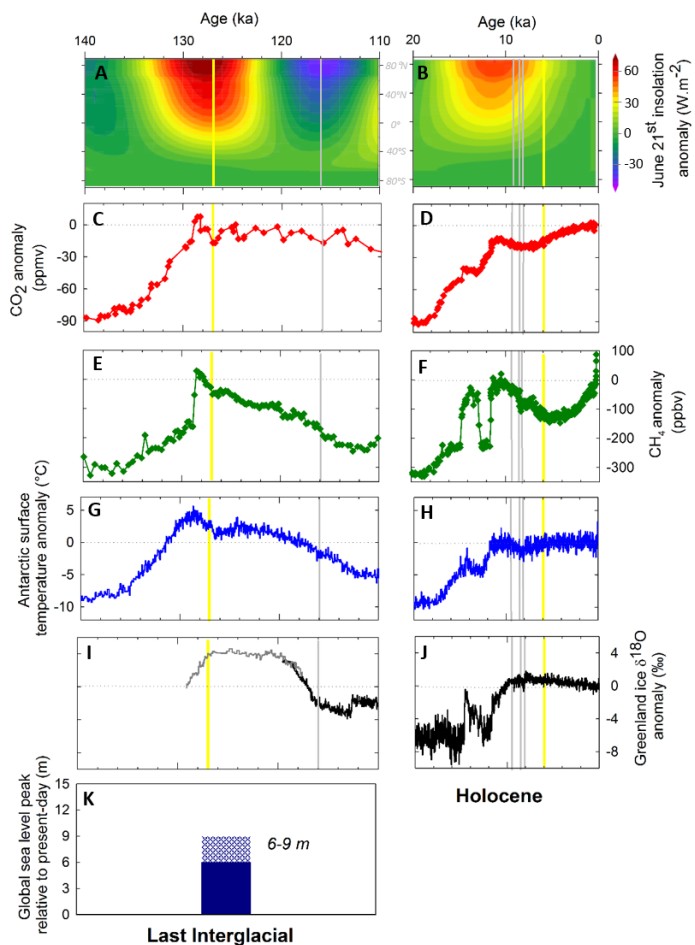

**Figure 1: Forcing and climatic records across the Last Interglacial (LIG, left) and the Holocene (right).**
Records are displayed in panels **A)** to **J)** as anomalies relative to their average value of the last 1000 years. **A** and **B)**
21st June insolation across latitudes; **C** and **D)** Atmospheric $CO_2$ concentration (Siegenthaler et al. 2005; Schneider
et al. 2013 CP, Monnin et al. 2004); **E** and **F)** Atmospheric $CH_4$ concentration (Loulergue et al. 2008); **G** and **H)**
Antarctic surface air temperature reconstruction (Jouzel et al. 2007); **I** and **J)** Greenland ice $\delta^{18}O$: from NEEM ice
core (NEEM community members 2012) in dark grey and from NGRIP ice core (NGRIP project members 2004) in
black. Note that NEEM ice $\delta^{18}O$ is shifted by +2‰. **K)** LIG maximum global mean sea level (GMSL) relative to
present-day, uncertainties in the amplitude are indicated by the shading (see Dutton et al. 2015 for a review). No
significant sea level variations are reported throughout the Holocene compared to present-day. NGRIP ice $\delta^{18}O$ is
displayed on the GICC05 annual layer-counted timescale (Svensson et al. 2008) over the last 20 ka and on the
AICC2012 chronology across the 119-110 ka time interval. All other ice core records are displayed on the
AICC2012 chronology which is coherent by construction with the GICC05 time scale over the last 60 ka (Bazin et
al. 2012, Veres et al. 2012). Vertical yellow lines indicate 127 and 6 ka, the time intervals chosen to run the
coordinated PMIP4-CMIP6 *lig127k* and *midHolocene* simulations. Vertical grey lines represent the 116, 9.5, 8.5 and
8.2 ka time intervals for which additional sensitivity simulations will be run within PMIP4.








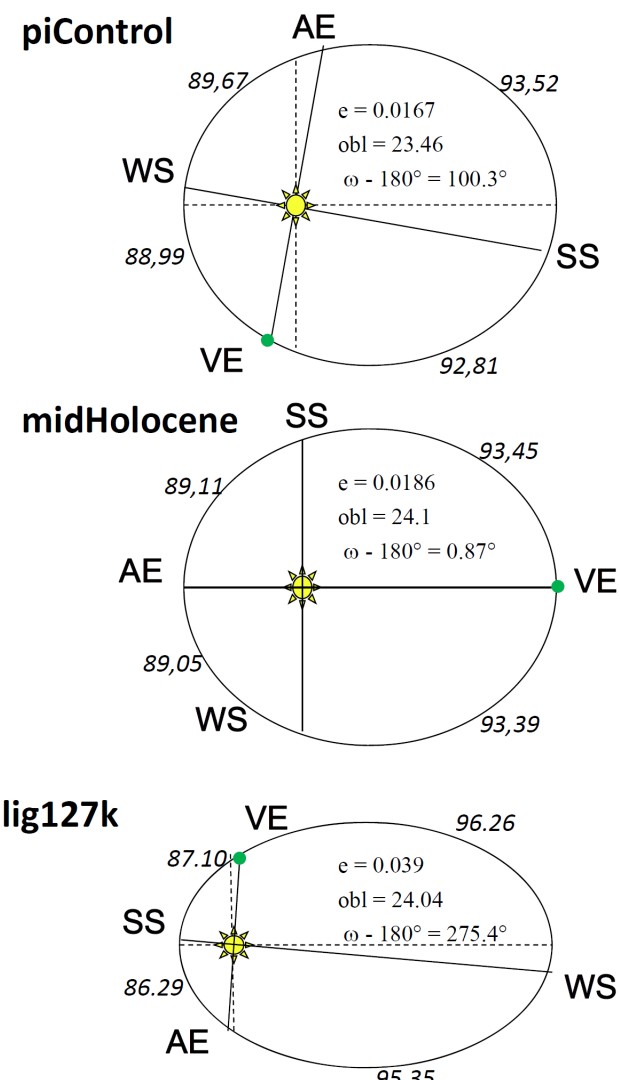

**Figure 2. Orbital configuration for *piControl, midHolocene, and lig127k* experiments.** Note that aspect ratio between the two axes of the ellipse has been magnified to better highlight the differences between the periods. However, the change in ratio between the different periods is proportional to the real values. In these graphs VE stands for vernal equinox, SS for summer solstice, AE for autumnal equinox, and WS for winter solstice. The numbers along the ellipse are the number of days between solstices and equinoxes.





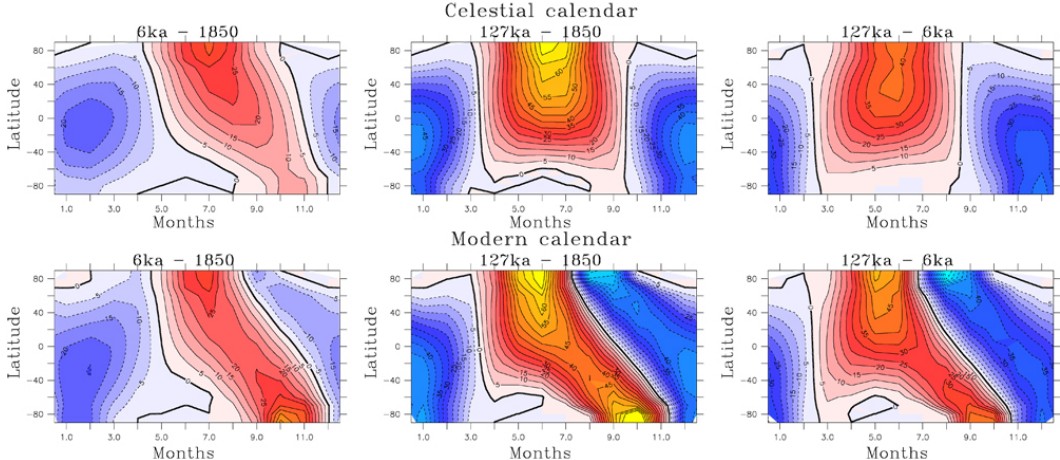

**Figure 3. Latitude-month insolation anomalies (127ka-1850, 6ka-1850, 127ka-6ka)** computed using either the celestial calendar (top) or the modern calendar (bottom), with vernal equinox on March 21 at noon, to compute monthly averages (W m$^{-2}$).


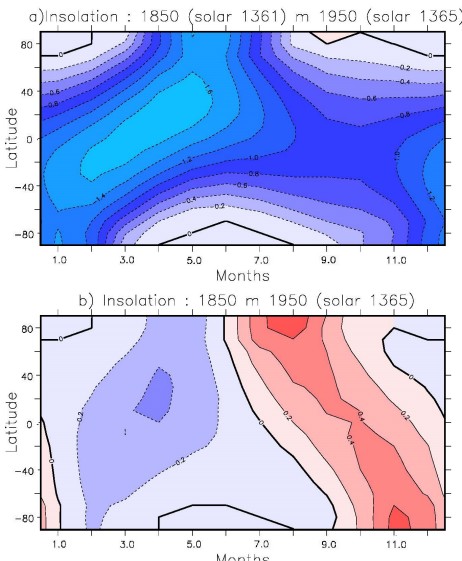

**Figure 4**. **Difference in incoming solar radiation at the top of the atmosphere (W m$^{-2}$) between PMIP4 and PMIP3 protocols**, a) considering the changes in Earth's orbital parameters between 1850 and 1950 and the reduction of the solar constant from 1365 to 1360.7 between these two PMIP phases and b) only the changes in Earth's orbital parameters.






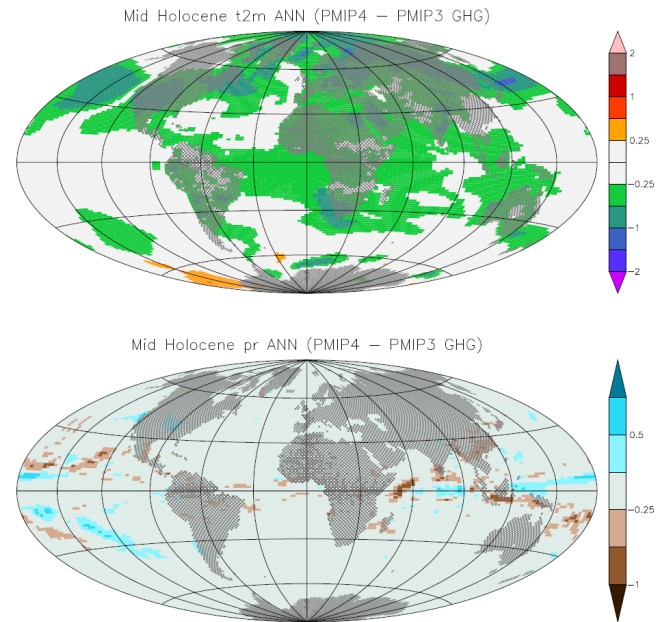

**Figure 5. Impact of the changes in trace gases between PMIP3 and PMIP4** on temperature (°C) and precipitation (mm d$^{-1}$) as estimated with the IPSLCM5A model. Only significant values are plotted in colors.







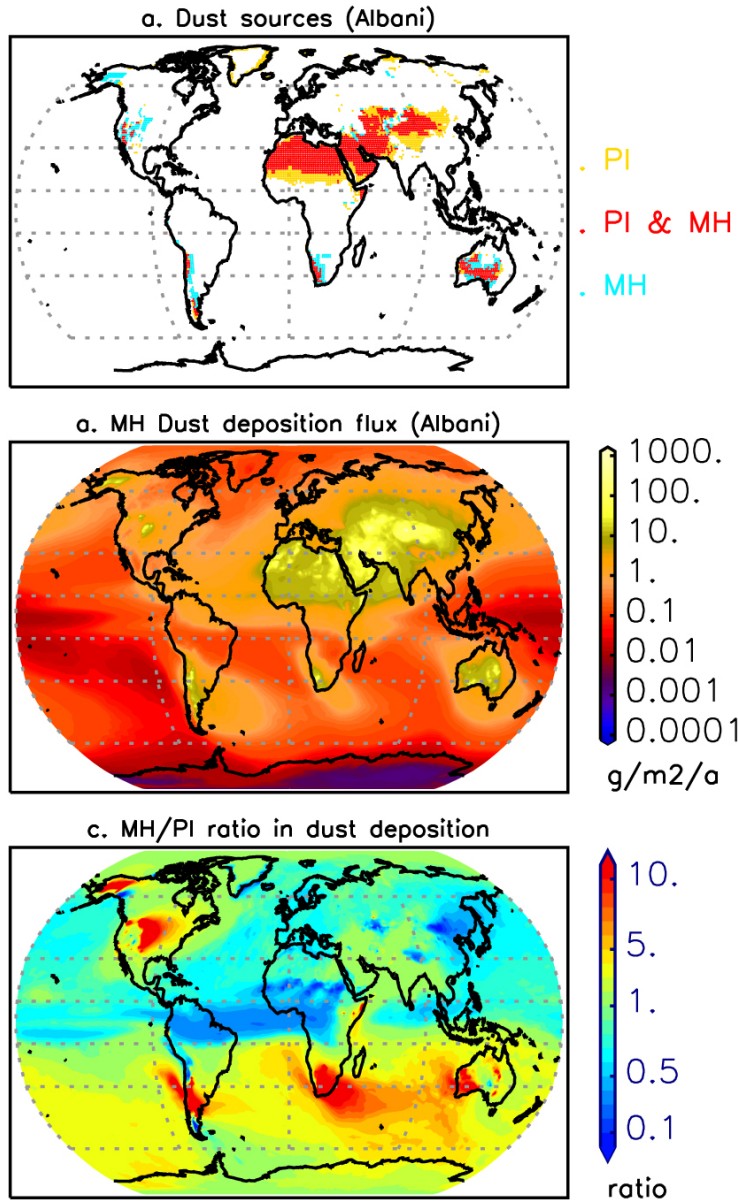

**Figure 6. Maps of dust from observationally-constrained simulations with the Community Climate System Model for the *midHolocene* (Albani et al., 2015).** a. Active sources for dust emissions for the *midHolocene* and the *piControl* (Albani et al., 2016). b. Dust deposition (g m$^{-2}$ a$^{-1}$) in the *midHolocene*. c. Ratio of *midHolocene* / *piControl* dust deposition.