# Peer review of "The PMIP4 contribution to CMIP6 - Part 2: Two Interglacials, Scientific Objective and Experimental Design for Holocene and Last Interglacial Simulations"

_Geoscientific Model Development, 2016_

## Referee Comment (RC1) · Anonymous Referee #1 · 9 Dec 2016

Review of manuscript entitled "The PMIP4 contribution to CMIP6 – Part 2: Two interglacials, scientific objective and experimental design for Holocene and Last Interglacial simulations."

This manuscript provides an extensive description of the experimental design of the two interglacial PMIP4/CMIP6 simulations. The manuscript is well written and provides modeling groups with all the details needed to perform one of the many described experiments. However, I do have a number of general comments and more technical notes that are listed below.

[Figure]

General comments:

I appreciate that fact that two main experiments have been defined, which also serve as entry cards for the other sensitivity tests. This will likely ensure a good number of groups participating in this MIP. However, when it comes to the sensitivity experiments, there seem to be many of them, and moreover, most of them provide a lot of freedom when it comes to the specifics of the experimental design. While I acknowledge that all the involved research groups have different foci, requiring a careful assessment of the proposed sensitivity experiments. However, it seems to me that in the current setup there are perhaps too many sensitivity experiments proposed, with too many options, with the likely result that in the end a proper model inter-comparison becomes difficult since very few groups in fact performed the same sensitivity experiments. The authors have certainly carefully considered the issue and thus the proposed simulations are possibly the optimal solution, however, it seems to me that the topic is of such importance that a general reviewer comment is justified.

The authors did not use phrases like 'future analogue' when describing the Holocene and Last Interglacial simulations, thereby carefully circumventing the discussion whether or not the lessons that one can learn from simulating these periods can be applied to future climate warming. Indeed they correctly describe that the main difference between these two periods and present-day is the latitudinal and seasonal distribution of incoming solar radiation (lines 83-86). However, it seems to me that this manuscript with this list of authors is the exact right spot to discuss this matter. Moreover, since in many studies this direct link between early Holocene / Last Interglacial and future climate change has been made and is still often made, not discussing it here can also be seen as a statement, be it a more concealed one. Therefor I strongly suggest to discuss the topic: What are the differences between early Holocene / Last Interglacial and future climate change? Which lessons from these past periods can be used to inform us about the future? Which systems and their sensitivity are influenced by the difference in the forcing and which ones are for instance solely driven by higher temper-

atures and can thus be seen as direct analogues? Do the authors have data available to provide some examples? Or have studies been done to investigate this? Possible examples that come to mind are Masson-Delmotte et al. (2006, DOI: 0.1007/s00382-005-0081-9c) and Blaschek et al. (2015, DOI: 10.1007/s00382-014-2279-1).

The manuscript describes quite extensively the proxy-based paleoclimate datasets that are available for the early Holocene and the Last Interglacial (section 4). However, it does not seem to come to any conclusions. Is this part solely meant to provide an overview? If so, is this the correct journal and manuscript to do so? Or do the authors mean to provide some guidance for future model-data inter-comparisons? In which case the conclusions of this section should be made more clear.

Minor and technical comments:

Lines 87-88: The second part of this sentence describes the possible results of these forcings, but since this paragraph solely describes the forcings themselves, it could be better to move this part to later in the manuscript.

Lines 87-88: Provide references for the warmer NH summers and enhanced NH monsoons.

Lines 106-107: Different model biases and the issue of model complexity are discussed here. How do the presented experimental designs overcome these issues? Please shortly discuss, perhaps later in the manuscript.

Lines 181-183: It would be easier for the reader if the order in which the two periods are described is always the same, either first early Holocene and then Last Interglacial or the other way around.

Line 197: What is meant with 'surface', really surface or something like 2m-temperature, reference temperature?

Line 198: Are these surface winds?

Line 284: Perhaps 130-126 ka for consistency.

Lines 344-348: Consider mentioning again that daily output is needed to calculate output on angular calendar months vs a fixed calendar months. Is the second part of this paragraph clear to the reader? Do they know what 'output needed to force regional area-limited models' is?

Lines 344-363: Are these paragraphs needed or can the manuscript simply refer to the website where all this information can be found?

Line 383: What feedbacks is this sentence referring to?

Line 404: 'can' or 'should' be prescribed from the last deglaciation experiment? It seems to me that these are the details that could in the end result in a model inter-comparison in which all simulations are slightly different from one another.

Lines 409-412: Is the CO2 concentration for 116ka so uncertain?

Lines 426-440: Make it more clear which of these sensitivity experiments are proposed for both the Holocene and the Last Interglacial and which ones only for the Holocene (in line with Table 2).

Line 445: remove first 'is'.

Line 445: year of reference to Hoelzmann is missing.

Lines 446-449: This part is rather vague. Are sensitivity experiments in this direction foreseen in PMIP4 or not?

Line 470: This line seems to suggest that the 'coral records' do not provide evidence of sea level rise. Please rephrase.

Line 471: Can this value also be given in Sv for easier comparison?

Line 484: 0.13Sv doesn't seem small at all, is this a typo?

Line 489: I understand that the 8.2ka-event happened close to 8.5ka, but is it really

necessary to introduce yet another simulation? Can't one simply use either 9.5 or 6ka in line with the other experiments?

Lines 489-492: This is somewhat unclear to me. Should the 2.5Sv pulse be introduced in year 500? Should both fluxes cease after this or should the background flux continue?

Lines 492-493: Does 'evidence for the recovery' mean that some upward trend should be visible or that it should again be close to the initial state? Why not give a more concrete number like a minimum of 100yrs after the end of the pulse?

Lines 479-493: It seems that this experimental design is similar to experiments that have been performed previously (Wagner et al., 2013, DOI: 10.1007/s00382-013-1706-z and Morrill et al., 2013, DOI: 10.5194/cp-9-955-2013), please refer to these manuscripts and discuss how and why those simulations differed from the design that is proposed here.

Lines 509-512: Why not start from 127ka? Should groups perform a 128ka equilibrium simulation as initial condition for the transient 128-122ka simulation? According to table 2 one should use 127ka as initial condition, but will this not lead to some spurious jump in the climate?

Line 540: Should this be 'small' rather then 'large'?

Line 572: Bakker et al. 2013 does not include LIG proxy-based climate reconstruction data. Perhaps Bakker et al., 2014 (DOI: 10.1016/j.quascirev.2014.06.031) is meant?

Lines 622-623: A number of datasets that are mentioned throughout the manuscript are not available on the website, when will they be?

Table 1: For 'Other GHG gases' 6ka and 127ka say '0', is that different from 'CMIP DECK piControl? What does 'SSI,ap if needed' mean?

Table 2: part of experiment 3.1 could also be considered as part of 3.3 (sensitivity to

ice sheets). The hol8.2 ka event simulation is somewhat confusing, should it be 8.2 or 8.5 ka orbital? Why is a freshwater forcing coming from the Antarctic Ice Sheet not taken into account?

Figure 1: Is the horizontal placement of the global sea-level peak in panel k suggesting the timing of the Last Interglacial high-stand?

Figures 3 and 4: Color bars are missing and quality is rather low.

---

## Referee Comment (RC2) · J. Brigham-Grette (Referee) · 28 Dec 2016

The paper is written as community vision for the next phase of coupled model experiments (CMIP) within the simulations guided by PMIP4. The long-standing project has successfully guided the community to test model sensitivity and process in understanding the role of forcing and feedbacks in the climate system. This project is a contribution to the WCRP Grand Science Challenges (stated at the end of the paper) with a focus on Earth system response to a variety of forcings, model system biases and how to assess future climate change given uncertainties in scenarios. The point of this paper

is to outline the planned experimental design criteria for assessing mid-Holocene (6 ka) and Last Interglacial (LIG, about 127ka) climate. The paper outlines requirements for everything from GHGs and orbital configurations to paleogeography and ice sheets, vegetation, aerosols. Requirements are also outlined for a set of Tier 2 experiments prescribing vegetation, ice sheets, meltwater sensitivity etc. A major focus of CMIP6 is on the realism of the mid-Holocene and LIG127 simulations compared to paleoclimate data (line 605).

My major complaint is that the modeling design does not really get at sea ice. Its mentioned a few times. It would seem absolutely necessary that different sea ice configurations are included the same way that different (or prescribed) ice sheet geographies are included. For 6k (with nearly ice free summers from 9 to 6ka? (Funder et al ) and for 127k (possible ice free summers at peak interglacial?; no sea ice south of Bering Strait, several papers) sea ice variability or 2-3 different modeled geographies might be considered.

The paper is well written and easy to follow however I had to read parts of the Eyring et al paper 2016 in this journal to find some of the terminology. I am not a modeler, yet I am among those in the community who would like to read about modeling project plans, but might not immediately understand what the "CMIP DECK" is. The paper does an impressive job listing summaries and paleodata compilations that might be used for input however it is not exhaustive.

There are a number of experiments described and the CMIP6 program includes 33 modeling groups (says website). So one would hope there are enough models and groups chasing each experiment for the models to be compared. Eyring et al 2016 says "To ensure community engagement, an important criterion was that enough modeling groups (at least eight) were willing to perform all of the MIP's highest priority (Tier 1) experiments and providing all the requested diagnostics needed to answer at least one of its leading science questions.

This paper should move forward to publication with only a few picky edits:

Line 110: I suggest for non-modelers that you add a footnote about what an "entry card" is? I understand this refers to a specific list of requirements. Line 129 : define ISMIP6 – Ice Sheet Model Intercomparison 6 contribution to CMIP.

Line 153: typo, Mollusc shells, not mullusc shells. Line 164: write out the meaning of DECK – Diagnostic, Evaluation and Characterization of Klima. One should not have to read the Eyring et al. 2016 paper to get all of the acronyms. Line 429: move the Lozhkin and Anderson reference. So it reads: . . ..vegetation and climate in which vegetation cover in the high-latitudes is changed from tundra to boreal forest (experiment a) (Lozhkin and Anderson, 1995) and the Sahara desert is replaced by evergreen shrub to 25°N and savanna/. . ..

Line 445: Add year to the Hoelzmann et al reference.

Line465 and section 3.4: Should/Could H11 be added to figure 1? Line 490: The location of the freshwater flux is extremely important and there might be reasons for the freshwater to hug the coast rather than be flooded over the entire Labrador sea. So this might also be part of the experimental design?

Line 1158: Fig. 1 caption. Add Veres et al, 2013 to the sentence containing AICC2012. Or perhaps better yet, cite the editors of the AICC2012 volume.

Lines 1175 and 1180: Add color bars to figure 3 and 4 because the print on the lines in the figure are very tiny.

---

## Referee Comment (RC3) · Anonymous Referee #3 · 2 Jan 2017

The authors present a comprehensive description of design and set up for simulations of the mid-Holocene climate at 6 ky BP and of the Last Interglacial (LIG) climate at 127 ky BP including a variety of sensitivity simulations. The paper is clearly written and up to the point. It is perfectly suited for publication in GMD.

I tend to disagree with Referee 1 that the paper should include some speculations on lessons learnt from the interglacial simulations for possible future climate change. The story of future climate change is about the response of our climate system to un-precedented strong variation in external forcing, while the mid-Holocene and the early

[Figure]

Eemian pose the challenge of explaining climate change in the presence of weak variation in external forcing. In any case, simulating the subtleties of past climate variability is a prerequisite for gaining confidence in understanding the dynamics of our climate system – which the authors clearly state.

Before publication, I would appreciate, if the authors could consider the following issues.

a) The authors correctly highlight uncertainties arising from prescribing or simulating Holocene and Eemian vegetation patterns. The authors recommend using the reconstruction by Hoelzmann et al. (1998) for Holocene North Africa. Is this still the best reconstruction? What about the reconstructions mentioned in the papers cited by the authors or by Lézine et al. (2011), Larrasoana et al. (2013) , . . .? Perhaps there are good reasons to still use Hoelzmann's et al data. But this should be critically reassessed.

b) In the same line: What about lakes? Lakes potentially matter in the mid-Holocene Sahara (e.g., Krinner et al., GRL, 2012). Perhaps also for lakes, the reconstruction by Hoelzmann et al. would be useful.

c) SST biases in the coupled atmosphere – ocean models presumably contribute to an underestimate of Interglacial Monsoon strengths. Hence some SST sensitivity experiments (e.g., strong vs weak gradient between tropical and extratropical SST differences between pre-industrial and mid-Holocene / LIG climate) might be worth considering.

d) Likewise, sensitivity experiments with respect to changes in Arctic sea-ice could be instructive to explore the role of high latitude climate system feedbacks (cf. the papers cited by the authors).

Minor issues:

i) The term Tier 1 (explained in line 132) should be defined earlier.

ii) Fig. 6: I do not understand what Figure 6a refers to. Is it just the geographic location

of dust sources, or the difference in locations – irrespective of their strength? Please use a superscript in the dimension g/m2/a .

iii) Fig. 5: What is the meaning of the different shades of grey on the continents? In the upper figure, the color grey also appears on the temperature scale (temperature difference between 1.5 and 2 K)?!

iv) Line 754 ff: Something went wrong with the citation of the Dahl Jensen and . . . and . . . and .

References to papers cited (and not included in the manuscript):

Krinner, G., Lezine, A. M., Braconnot, P., Sepulchre, P., Ramstein, G., Grenier, C., & Gouttevin, I. 2012). A reassessment of lake and wetland feedbacks on the North African Holocene climate. Geophysical Research Letters, 39. doi:10.1029/2012gl050992

Larrasoana JC, Roberts AP, Rohling EJ (2013) Dynamics of Green Sahara Periods and Their Role in Hominin Evolution. PLoS ONE 8(10): e76514.doi:10.1371/journal.pone.0076514

Lézine, A. M., Zheng, W., Braconnot, P., & Krinner, G. (2011). Late Holocene plant and climate evolution at Lake Yoa, northern Chad: pollen data and climate simulations. Climate of the Past, 7 (4), 1351-1362.

---

## Referee Comment (RC4) · Anonymous Referee #1 · 2 Jan 2017

Additional note from referee #1

As a follow-up on the interactive comment from referee #3, I would like to add some clarifying words.

Past interglacials, especially the early Last Interglacial, are often regarded as analogues for future climate change. Whether this is justified or not is an open question and I'm not asking the authors to provide any proof in favour or against this idea. However, it seems to me that the authors have made a well considered choice not to mention the Last Interglacial (or the Early Holocene for that matter) in this context and it

would be highly interesting for the larger community, especially considering the wealth of knowledge on the topic held within the long list of authors, if these considerations would be part of the manuscript.

―――――――――――――――――――

---

## Author Response (AR1)

This document includes responses (in blue) to all the Reviewer comments.

Response to Interactive comments by Anonymous Referee #1 [RC1]

However, it seems to me that in the current setup there are perhaps too many sensitivity experiments proposed, with too many options, with the likely result that in the end a proper model inter-comparison becomes difficult since very few groups in fact performed the same sensitivity experiments. The authors have certainly carefully considered the issue and thus the proposed simulations are possibly the optimal solution, however, it seems to me that the topic is of such importance that a general reviewer comment is justified.

*We agree. With so many sensitivity expts there is the possibility of too few modeling groups doing the same experiments. We have limited the lig116k glacial inception to the reconstructed value of CO2 and kept only the prescribed boreal forest and shrub savanna experiments to test the sensitivities to more idealized vegetation configurations. We also now suggest that the hol8.2k experiment can start from the hol9.5k simulation.*

Therefore I strongly suggest to discuss the topic: What are the differences between early Holocene / Last Interglacial and future climate change? Which lessons from these past periods can be used to inform us about the future? Which systems and their sensitivity are influenced by the difference in the forcing and which ones are for instance solely driven by higher temperatures and can thus be seen as direct analogues? Do the authors have data available to provide some examples? Or have studies been done to investigate this? Possible examples that come to mind are Masson-Delmotte et al. (2006, DOI: 0.1007/s00382-005-0081-9c) and Blaschek et al. (2015, DOI: 10.1007/s00382-014-2279-1).

*We have added a short discussion to the Conclusions. Neither time period can be suggested as a true paleo-analogue for the future because of the seasonal nature of the orbital forcing. That said, higher temperatures in the polar regions, particularly during the summer months and for the Last Interglacial, directly influence sea ice and the ice sheets. The data evidence provides a means of evaluating if we are capturing this sensitivity correctly in models being used for projections of future climate change.*

The manuscript describes quite extensively the proxy-based paleoclimate datasets that are available for the early Holocene and the Last Interglacial (section 4). However, it does not seem to come to any conclusions. Is this part solely meant to provide an overview? If so, is this the correct journal and manuscript to do so? Or do the authors mean to provide some guidance for future model-data inter-comparisons? In which case the conclusions of this section should be made more clear.

*The goal here was to summarise the data sets that are available for different types of model evaluation for the MH and LIG, and thus we explicitly draw attention to the state-of-the-art syntheses of data on e.g. hydrology, vegetation, climate reconstruction, climate variability etc in this section. We feel that it is important to do this in the current paper (a) to make it clear that model evaluation of the experiments is feasible, and (b) so that modelers are aware of the available data sets and their limitations. However, we agree that it would be useful to add something at the end of this section in conclusion. So we have added the following:*

*"The public-access data sets currently available for the MH and LIG serve different functions and address different aspects of the climate system. Modeling groups running MH and LIG simulations, or sensitivity experiments, are encouraged to work with data experts, using multiple data sets for a full diagnosis of the simulations. Many of these data sets provide measures of the uncertainty of the reconstructions and data-model comparisons should be designed to take these uncertainties into account."*

Minor and technical comments:

Lines 87-88:  The second part of this sentence describes the possible results of these forcings, but since this paragraph solely describes the forcings themselves, it could be better to move this part to later in the manuscript.
*Moved to next paragraph.*

Lines 87-88:  Provide references for the warmer NH summers and enhanced NH monsoons.
*References have been added.*

Lines 106-107:  Different model biases and the issue of model complexity are discussed here. How do the presented experimental designs overcome these issues?  Please shortly discuss, perhaps later in the manuscript.
*Deleted sentence.*

Lines 181-183:  It would be easier for the reader if the order in which the two periods are described is always the same, either first early Holocene and then Last Interglacial or the other way around.
*Order switched here to be consistent with discussion of other forcings*

Line 197: What  is meant  with  'surface', really surface or something like 2m- temperature, reference temperature?
Line 198: Are these surface winds?
*We have moved this paragraph and discussion to the Section 6 on Data Availability and made consistent with the text in the revised version of the CMIP6-PMIP4 Overview paper in GMDD*

Line 284: Perhaps 130-126 ka for consistency.
*Revised as suggested.*

Lines 344-348:  Consider mentioning again that daily output is needed to calculate output on angular calendar months vs a fixed calendar months. Is the second part of this paragraph clear to the reader? Do they know what 'output needed to force regional area-limited models' is?
*We have made consistent with the text in the revised version of the CMIP6-PMIP4 Overview paper in GMDD. This text now appears in Section 6 on Data Availability of our paper. A supplementary information table has been added to the paper and also a link to the PMIP4 web site.*

Lines 344-363: Are these paragraphs needed or can the manuscript simply refer to the website where all this information can be found?
*We have not shortened this section on the advice of GMD Chief Editor J. Hargreaves. We have added a list of variables to be saved in the SI.*

Line 383: What feedbacks is this sentence referring to?
*Albedo and freshwater*

Line 404: 'can' or 'should' be prescribed from the last deglaciation experiment? It seems to me that these are the details that could in the end result in a model inter- comparison in which all simulations are slightly different from one another.
*Should*

Lines 409-412: Is the $CO_2$ concentration for 116ka so uncertain?
*The Tier 2 experiment will use the EPICA Dome C data published by Schneider et al., 2013 CP (that is also publishing d13-$CO_2$ for the same depth intervals) which is the currently best $CO_2$ data set for the Eemian. For the 116 ka BP, the nearest data point at 115909 ka BP (on AICC12) has 273 ppm in $CO_2$.*

Lines 426-440: Make it more clear which of these sensitivity experiments are proposed for both the Holocene and the Last Interglacial and which ones only for the Holocene (in line with Table 2).
*Revised as suggested.*

Line 445: remove first 'is'.
*Paragraph deleted.*

Line 445: year of reference to Hoelzmann is missing.
Paragraph deleted.

Lines 446-449: This part is rather vague. Are sensitivity experiments in this direction foreseen in PMIP4 or not?
*Paragraph deleted.*

Line 470: This line seems to suggest that the 'coral records' do not provide evidence of sea level rise. Please rephrase.
*Clarified*

Line 471: Can this value also be given in Sv for easier comparison?
*Done*

Line 484: 0.13Sv doesn't seem small at all, is this typo?
*Revised to be clearer*

Line 489:  I understand that the 8.2ka-event happened close to 8.5ka, but is it really necessary to introduce yet another simulation? Can't one simply use either 9.5 or 6ka in line with the other experiments?

*We agree. This proposed experiment now starts from a 9.5 ka simulation.*

Lines 489-492: This is somewhat unclear to me.   Should the 2.5Sv pulse be intro- duced in year 500?  Should both fluxes cease after this or should the background flux continue?

*We agree that as first written the experimental protocol was unclear. Revised to be clearer.*

Lines 492-493:  Does 'evidence for the recovery' mean that some upward trend should be visible or that it should again be close to the initial state?  Why not give a more concrete number like a minimum of 100yrs after the end of the pulse?

*We agree. Sentence revised.*

Lines 479-493: It seems that this experimental design is similar to experiments that have been performed previously  (Wagner   et al., 2013, DOI: 10.1007/s00382-013-1706-z and Morrill et al., 2013, DOI: 10.5194/cp-9-955-2013), please refer to these manuscripts and discuss how and why those simulations differed from the design that is proposed here.

*We have revised the experimental design for the freshwater inputs to be the 'Lake + Ice_100 yrs' scenario of Wagner et al. (2013). As they suggest this design is more consistent with ice dynamics and the data of Carlson et al. (2009) than the shorter 1-yr flood scenarios described in Morrill et al., 2013.*

Lines 509-512:  Why not start from 127ka? Should groups perform a 128ka equilibrium simulation as initial condition for the transient 128-122ka simulation? According to table 2 one should use 127ka as initial condition, but will this not lead to some spurious jump in the climate?

*The orbital and GHG forcings are similar for 128 ka and 127 ka but agree that a more consistent experimental protocol is to start from 127 ka. Ending date also revised to 121 ka to be of similar length as midHolocene transient simulation.*

Line 540: Should this be 'small' rather then 'large'?

*Yes, small is correct. We corrected.*

Line 572:  Bakker et al. 2013 does not include LIG proxy-based climate reconstruction data. Perhaps Bakker et al., 2014 (DOI: 10.1016/j.quascirev.2014.06.031) is meant?

*We originally included this because it uses climate reconstructions to evaluate the simulations, including simulation of sea ice and high-latitude temperatures. We have now taken the reference out. Bakker et al (2014) is also a data-model comparison paper, rather than a primary source for reconstructions.*

Lines 622-623: A number of datasets that are mentioned throughout the manuscript are not available on the website, when will they be?

*Datasets are now available on the website*

Table 1: For 'Other GHG gases' 6ka and 127ka say '0', is that different from 'CMIP DECK piControl?

*Yes, there are no CFCs included in the midHolocene and lig127k simulations.*

What does 'SSI,ap if needed' mean?

*Deleted. Refers to the global level of geomagnetic activity, and are used as inputs to parameterisations of magnetospheric particle precipitation.*

Table 2: part of experiment 3.1 could also be considered as part of 3.3 (sensitivity to ice sheets).

*Yes, but we have decided to keep separate.*

The hol8.2 ka event simulation is somewhat confusing, should it be 8.2 or 8.5 ka orbital?

*We now suggest that the hol8.2k experiment should start from the hol9.5k simulation. Except for the freshwater forcing, all other forcings and boundary conditions remain the same as the hol9.5k simulation.*

Why is a freshwater forcing coming from the Antarctic Ice Sheet not taken into account?

*The early last interglacial freshwater forcing associated with the H11 event is likely to have come from the melting of the MIS6 NH ice sheets. Reconstructions from marine data support this.*

Figure 1: Is the horizontal placement of the global sea-level peak in panel k suggesting the timing of the Last Interglacial high-stand?

*No. We will clarify this in the legend.*

Figures 3 and 4: Color bars are missing and quality is rather low.

*Redrafted to improve quality. Color bars added.*

Response to Interactive comments by Julie Brigham-Grette [RC2]

My major complaint is that the modeling design does not really get at sea ice. Its mentioned a few times. It would seem absolutely necessary that different sea ice configu- rations are included the same way that different (or prescribed) ice sheet geographies are included. For 6k (with nearly ice free summers from 9 to 6ka? (Funder et al) and for 127k (possible ice free summers at peak interglacial?; no sea ice south of Bering Strait, several papers) sea ice variability or 2-3 different modeled geographies might be considered.

*All the sensitivity simulations proposed are to use the same coupled atmosphere-ocean-sea ice models as the midHolocene and lig127k CMIP6 or PMIP4 simulations to assess missing forcings/boundary conditions that affect the coupled climate state. Although AMIP simulations with different scenarios of seasonal sea ice extent could be of interest, we do not include here.*

The paper is well written and easy to follow however I had to read parts of the Eyring et al paper 2016 in this journal to find some of the terminology. I am not a modeler, yet I am among those in the community who would like to read about modeling project plans, but might not immediately

understand what the "CMIP DECK" is. The paper does an impressive job listing summaries and paleodata compilations that might be used for input however it is not exhaustive.

This paper should move forward to publication with only a few picky edits:

Line 110: I suggest for non-modelers that you add a footnote about what an "entry card" is? I understand this refers to a specific list of requirements.
*As indicated in subsequent text: "all modeling groups contributing to PMIP4-CMIP6 must perform either the midHolocene experiment or a simulation of the Last Glacial Maximum (Kageyama et al., 2016).*

Line 129: define ISMIP6 – Ice Sheet Model Intercomparison 6 contribution to CMIP.
*Defined.*

Line 153: typo, Mollusc shells, not mullusc shells.
*Corrected*

Line 164: write out the meaning of DECK – Diagnostic, Evaluation and Characterization of Klima. One should not have to read the Eyring et al. 2016 paper to get all of the acronyms
*We have revised as suggested.*

.Line 429: move the Lozhkin and Anderson reference. So it reads: . . .. vegetation and climate in which vegetation cover in the high-latitudes is changed from tundra to boreal forest (experiment a) (Lozhkin and Anderson, 1995) and the Sahara desert is replaced by evergreen shrub to 25◦N and savanna/. . ..
*Paragraph deleted.*

Line 445: Add year to the Hoelzmann et al reference.
*Paragraph deleted.*

Line465 and section 3.4: Should/Could H11 be added to figure 1? Line 490: The location of the freshwater flux is extremely important and there might be reasons for the freshwater to hug the coast rather than be flooded over the entire Labrador sea. So this might also be part of the experimental design?
*We have removed all gray lines from Figure 1. We propose starting the H11 experiment from the 127ka simulation. True, this is an idealization but more feasible than adding an additional time slice simulation for some time between 132-130ka. This should give the first-order approximation of the climate state at 130ka. Again to make this simulation feasible for many modeling groups, we have suggested for this Tier 2 simulation to just add the freshwater across the North Atlantic. Although the European ice sheet is reconstructed to be larger in MIS6 than MIS2, the partitioning of the H11 freshwater between the European and Laurentide ice sheets is not clear.*

Line 1158: Fig. 1 caption. Add Veres et al, 2013 to the sentence containing AICC2012. Or perhaps better yet, cite the editors of the AICC2012 volume.

*Veres et al, 2013 moved as suggested.*

Lines 1175 and 1180:  Add color bars to figure 3 and 4 because the print on the lines in the figure are very tiny.
*Done.*

Response to Interactive comments by Anonymous Referee #3 [RC3]

I tend to disagree with Referee 1 that the paper should include some speculations on lessons learnt from the interglacial simulations for possible future climate change. The story of future climate change is about the response of our climate system to unprecedented strong variation in external forcing, while the mid-Holocene and the early Eemian pose the challenge of explaining climate change in the presence of weak variation in external forcing. In any case, simulating the subtleties of past climate variability is a prerequisite for gaining confidence in understanding the dynamics of our climate system – which the authors clearly state.

Before publication, I would appreciate, if the authors could consider the following issues.
a) The authors correctly highlight uncertainties arising from prescribing or simulating Holocene and Eemian vegetation patterns. The authors recommend using the reconstruction by Hoelzmann et al. (1998) for Holocene North Africa. Is this still the best reconstruction? What about the reconstructions mentioned in the papers cited by the authors or by Lézine et al. (2011), Larrasoana et al. (2013) , : : :? Perhaps there are good reasons to still use Hoelzmann's et al data. But this should be critically reassessed.
*We agree with Reviewer #1 that there are perhaps too many sensitivity experiments proposed, with too many options.  With so many sensitivity expts there is the possibility of too few modeling groups doing the same experiments. We have kept only the prescribed boreal forest and shrub savanna experiments to test the sensitivities to more idealized vegetation, consistent with reconstructions, among models and for comparing these two time periods.*

b) In the same line: What about lakes? Lakes potentially matter in the mid-Holocene Sahara (e.g., Krinner et al., GRL, 2012). Perhaps also for lakes, the reconstruction by Hoelzmann et al. would be useful.
*We agree that lakes could potentially impact the monsoon climate, and there were a number of papers showing this prior to the Krinner et al (2012) paper. Indeed, the Hoelzmann et al (1998) data set was explicitly constructed to allow such sensitivity experiments to be made, see Broström, A., Coe, M., Harrison, S.P., Gallimore, R., Kutzbach, J.E., Foley, J., Prentice, I.C. and Behling, P., 1998. Land surface feedbacks and palaeomonsoons in northern Africa. Geophysical Research Letters 25: 3615-3618. However, previous work suggests that the vegetation changes appear to be the most important element of the land-surface feedback – and given limited resources, our focus in PMIP and in this paper is therefore on designing common experiments to examine the sensitivity to vegetation.*

c) SST biases in the coupled atmosphere – ocean models presumably contribute to an

underestimate of Interglacial Monsoon strengths. Hence some SST sensitivity experiments (e.g., strong vs weak gradient between tropical and extratropical SST differences between pre-industrial and mid-Holocene / LIG climate) might be worth considering.

*All the sensitivity simulations proposed are to use the same coupled atmosphere-ocean-sea ice models as the midHolocene and lig127k CMIP6 or PMIP4 simulations to assess missing forcings/boundary conditions that affect the coupled climate state. Although AMIP simulations with different gradients of SST or scenarios of seasonal sea ice extent could be of interest, we do not include here.*

d) Likewise, sensitivity experiments with respect to changes in Arctic sea-ice could be instructive to explore the role of high latitude climate system feedbacks (cf. the papers cited by the authors).
*See comment above.*

Minor issues:
i) The term Tier 1 (explained in line 132) should be defined earlier.
*Defined.*

ii) Fig. 6: I do not understand what Figure 6a refers to. Is it just the geographic location of dust sources, or the difference in locations – irrespective of their strength? Please use a superscript in the dimension g/m2/a .
*Revised.*

iii) Fig. 5: What is the meaning of the different shades of grey on the continents? In the upper figure, the color grey also appears on the temperature scale (temperature difference between 1.5 and 2 K)?!
*We agree that the color scales are confusing in this figure. This figure has been redrafted.*

iv) Line 754 ff: Something went wrong with the citation of the Dahl Jensen and : : : and : : : and .
*Corrected.*

References to papers cited (and not included in the manuscript):
Krinner, G., Lezine, A. M., Braconnot, P., Sepulchre, P., Ramstein, G., Grenier, C., & Gouttevin, I. 2012). A reassessment of lake and wetland feedbacks on the North African Holocene climate. Geophysical Research Letters, 39. doi:10.1029/2012gl050992.

Larrasoana JC, Roberts AP, Rohling EJ (2013) Dynamics of Green Sahara Periods and Their Role in Hominin Evolution. PLoS ONE 8(10): e76514.doi:10.1371/journal.pone.0076514

Lézine, A. M., Zheng, W., Braconnot, P., & Krinner, G. (2011). Late Holocene plant and climate evolution at Lake Yoa, northern Chad: pollen data and climate simulations. Climate of the Past, 7 (4), 1351-1362.
*Now included.*

Additional note from referee #1 [RC4]

As a follow-up on the interactive comment from referee #3, I would like to add some clarifying words.

Past interglacials, especially the early Last Interglacial, are often regarded as analogues for future climate change. Whether this is justified or not is an open question and I'm not asking the authors to provide any proof in favour or against this idea. However, it seems to me that the authors have made a well-considered choice not to mention the Last Interglacial (or the Early Holocene for that matter) in this context and it
Would be highly interesting for the larger community, especially considering the wealth of knowledge on the topic held within the long list of authors, if these considerations would be part of the manuscript
*See response to Referee 1 point 2. We have added some discussion to the Conclusions.*

[revised manuscript text omitted]

**piControl**

AE

*89,67*  *93,52*

e = 0.0167
obl = 23.46
ω - 180° = 100.3°

WS

SS

*88,99*

VE

*92,81*

**midHolocene**

SS

*89,11*  *93,45*

e = 0.0186
obl = 24.1
ω - 180° = 0.87°

AE

VE

*89,05*

WS

*93,39*

**lig127k**

VE

*87.10*  *96.26*

e = 0.039
obl = 24.04
ω - 180° = 275.4°

SS

WS

*86.29*

AE

*95.35*

**Figure 2. Orbital configurations for *piControl, midHolocene, and lig127k* experiments.** Note that aspect ratio between the two axes of the ellipse has been magnified to better highlight the differences between the periods. However, the change in ratio between the different periods is proportional to the real values. In these graphs VE stands for vernal equinox, SS for summer solstice, AE for autumnal equinox, and WS for winter solstice. The numbers along the ellipse are the number of days between solstices and equinoxes.

1525

[Figure]

[Figure]

**Figure 3. Latitude-month insolation anomalies (6ka-1850, 127ka-1850, 127ka-6ka)** computed using either the celestial calendar (top) or the modern calendar (bottom), with vernal equinox on March 21 at noon, to compute monthly averages (W m$^{-2}$).

[Figure]

1565     **Figure 4. Difference in incoming solar radiation at the top of the atmosphere (W m$^{-2}$) between PMIP4 and PMIP3 protocols**, a) considering the changes in Earth's orbital parameters between 1850 and 1950 and the reduction of the solar constant from 1365 to 1360.7 between these two PMIP phases and b) only the changes in Earth's orbital parameters.

[Figure]

**Figure 5. Impact of the changes in trace gases specified for 6 ka between PMIP3 and PMIP4 on surface air** temperature (°C) and precipitation (mm d⁻¹) as estimated with the IPSLCM5A model. Only significant values are plotted in colors.

[Figure]

**Figure 6. Maps of dust from observationally-constrained simulations with the Community Climate System Model for the *midHolocene* (Albani et al., 2015).** a. Active sources for dust emissions for the *midHolocene* and the *piControl* (Albani et al., 2016). b. Dust deposition (g m⁻² a⁻¹) in the *midHolocene*. c. Ratio of *midHolocene* / *piControl* dust deposition.

[Figure]

Appendix A: Variable request
This list represents what is currently available from the official CMIP6 source
(http://proj.badc.rl.ac.uk/svn/exarch/CMIP6dreq/tags/latest/dreqPy/docs/CMIP6_MIP_tables.xlsx).

For updates, users should refer to the website with the PMIP data request
(https://pmip4.lsce.ipsl.fr/doku.php/database:pmip4request#the_pmip4_request).

| Page 10: [1] Deleted | Bette Otto-Bliesner | 3/21/17 12:33:00 PM |
|---|---|---|

The simulations should follow the CMIP6 data request and format. For groups only contributing to PMIP, the data format and organization on the ESGF archive is the same as for CMIP6, except that the provision of daily values can be limited to 2D surface variables, including temperature, precipitation and winds. Groups are also asked to keep a 20 year period with all the output needed to force regional area-limited models, since we would like to strengthen the linkages between global and regional simulations for regional model-data comparisons.

The difference in orbital configuration between 127 ka, 6 ka and preindustrial means that there are differences in season length that should be accounted for in calculating seasonal changes (Kutzbach and Gallimore, 1988). The bias introduced from using the modern calendar rather than the celestial calendar to calculate seasonal averages is shown in Fig. 2, when the date of the vernal equinox is assigned to March 21 at noon. To be able to account for this effect when comparing the simulations to the paleoclimate reconstructions, daily outputs of at least surface temperature, precipitation and winds must be archived. Programs that provide an approximate estimate of monthly means on the fixed-angular celestial calendar from fixed-day calendar will be available on the PMIP4 web page.

| Page 32: [2] Deleted | Bette Otto-Bliesner | 3/16/17 7:38:00 PM |
|---|---|---|

CMIP DECK *piControl*

| Page 32: [3] Deleted | Bette Otto-Bliesner | 3/16/17 7:36:00 PM |
|---|---|---|

SSI, ap if needed

| Page 33: [4] Deleted | Bette Otto-Bliesner | 3/19/17 3:06:00 PM |
|---|---|---|

**PMIP4-CMIP6 Tier 1 simulations**

| | Entry card: *midHolocene* | *lig127k* |
|---|---|---|

| Page 33: [5] Deleted | Bette Otto-Bliesner | 3/5/17 2:41:00 PM |
|---|---|---|

vegetation reconstructions[2]
    *midHolocene* equilibrium veg with dgvm in *piControl*
.

| Page 33: [6] Deleted | Bette Otto-Bliesner | 3/5/17 2:42:00 PM |
|---|---|---|

Orbital: 8.2 ka
Ice sheet: ICE-6G or GLAC-1D reconstruction[1]
GHG: same as for the deglaciation experiment[1]
Initial state*:* 8.5 ka simulation

| Page 34: [7] Deleted | Bette Otto-Bliesner | 3/5/17 2:45:00 PM |
|---|---|---|

[2]Hoelzmann et al., 1998; Bigelow et al., 2003; available on the PMIP4 web page

[Figure]

Age (ka)

A  B

June 21st insolation anomaly (W.m⁻²)

$CO_2$ anomaly (ppmv)

C  D

$CH_4$ anomaly (ppbv)

E  F

Antarctic surface temperature anomaly (°C)

G  H

I  J

Greenland ice δ18O anomaly (‰)

Holocene

K

Global sea level peak relative to present-day (m)

6-9 m

Last Interglacial

---

## Author Response (AR2)

**Response to GMD Editors' comments on "The PMIP4 contribution to CMIP6 - Part 2: Two Interglacials, Scientific Objective and Experimental Design for Holocene and Last Interglacial Simulations" by Otto-Bliesner et al.**

Thank you for your patience on our resubmission of our manuscript addressing the Editors' comments. Indeed, the revisions to the comments were not major though it took a bit of time to ensure that the three PMIP4 papers describing the Last Millennium, Interglacials, and Last Glacial Maximum similarly handled the description of the "Tiers", documentation, and data distribution. Working together, Masa, Johann, and I have coordinated our responses to the concerns of the Editors and have revised our papers to be more consistent.

For clarity, we reproduce the Topical Editor and other Editors' comments in *blue/italic* and provide answers in black. Changes to the manuscript are presented in **bold face**.

First, addressing Topical Editor comments:

1. *The second response to reviewer #3 does not seem to reflect on the question, is that a copy/paste mistake?*

Not sure what response is being referred to. Our responses to Reviewer #3, parts a-d seem appropriate. Please advise further.

2. *In the Abstract and beginning of the manuscript you are relating all the time to "Tier 1" simulations. However, these are only defined on page 4 of the current version of the manuscript. Please reword to avoid this. This also connects to the general comment sent earlier through Julia Hargreaves.*

The reference to "Tier 1" in the abstract has been deleted. We now include a new second paragraph to the Introduction (lines 80+ in the tracked changes version) describing the Tier structure. See response to #6 below for more details and new text.

3. *Page 8, line 306 "Antarctic ice-sheets is" singular/plural form?*

Corrected to be singular form.

4. *I do not understand the sentence on page 11, first line "Albedo ...forcing"*

The sentence is indeed confusing. As such, we have decided to delete this sentence.

5. *I have the same particular comments than Julia Hargreaves on the ESGF distribution and standards in your paragraph 6. Also the same issue in specifying the conditions for checking equilibrium which are vague and difficult to find in the present manuscript.*

See responses to #7-10 below.
* * *
Second, addressing the general comments by the Editors of the three PMIP4 papers:

6. *Some papers mention the entry cards for CMIP, but others do not. Some bring in the "Tier" nomenclature without defining it, while the LGM discusses "sensitivity" experiments, leaving it uncertain as to how these fit into the CMIP system.*

   *\* All papers should explain the entry card system and contain full and consistent information on the documentation requirement for the runs. Many groups will do all the Tier 1 runs, so it really will be very confusing if the documentation requirements are different! All the papers should make sure to define CMIP concepts such as "Tier 1".*

We have now categorized all experiments in the PMIP4 papers as Tier 1, Tier 2 and Tier 3. The concept of Tier is introduced in the CMIP6 overview paper (Eyring et al., 2016), with Tier 1 having the highest priority. Within PMIP4, the Tier 1 experiments are those, which are Tier 1 for CMIP6 as well. They are also reference experiments for Tier 2 and Tier 3 experiments described for each period in the PMIP4 GMD manuscripts, which will be made clearer in the revised versions (especially for the LGM sensitivity experiments, which will be categorized as Tier 2). In the PMIP4-CMIP6 overview paper, we introduce the concept of PMIP4 entry card, capitalizing on PMIP's previous experiments. These (the *midHolocene* and *lgm* experiments) are special PMIP4-CMIP6 Tier 1 experiments. We consider that at least one of these experiments must be performed by the modelling groups to be part of PMIP4-CMIP6, because they will allow us to monitor the progress from the previous phases of PMIP and CMIP. This is explained in the overview paper, but will be recalled in the PMIP4 papers. We want to make it clear that the Tier 2 and 3 experiments absolutely require the corresponding Tier 1 experiment for their analysis, so the groups must perform the Tier 1 experiment first.

In the Interglacial manuscript we have included a new second paragraph to the Introduction:

**This paper is part of a suite of five manuscripts documenting the PMIP4 contributions to CMIP6. Kageyama et al. (2016) provide an overview on the five selected time periods and the experiments. More specific information is given in the contributions for the last millennium (*past1000*) by Jungclaus et al. (2016), for the last glacial maximum (*lgm*) by Kageyama et al. (2017), for the mid-Pliocene warm period (*midPliocene-eoi400*) by Haywood et al. (2016), and the present manuscript mid-Holocene (*midHolocene*) and the previous interglacial (*lig127k*). PMIP4 has adopted the CMIP6 categorization where the highest-priority experiments are classified as Tier 1, whereas additional sensitivity experiments or dedicated studies are Tier 2 or Tier 3. The standard experiments for the five periods are all ranked Tier 1. Tier 2 and 3 experiments absolutely require the corresponding Tier 1 experiment for their analysis, so the groups must perform the Tier 1 experiment first. Modelling groups are not obliged to run all PMIP4-CMIP6 experiments. It is mandatory, however, for all participating groups to run at least one of the experiments that were run in previous phases of PMIP (i.e., *midHolocene* or *lgm*).**

7. *The documentation requirement seems to be different in all the papers (and the LGM is actually inconsistent with the overview paper). The documentation requirement is really presented as part of the protocol, so should be included in each of the individual papers.*

The overview paper will insist on the importance of documenting the simulations. The specificities for each simulation are detailed in each paper, since they depend on the forcings for each experiment. This includes the documentation on spin-up and equilibrium as detailed below.

We have streamlined the documentation requirements and each manuscript now contains a section "Documentation". The specifications are, of course, different for the different time periods and experiments.

**2.6 Setup and documentation of simulations**

**To provide initial conditions for the simulations, it is recommended that a spin-up simulation is performed departing from the CMIP6 *piControl* experiment. The length of this spin-up simulation will be model- and resource- dependent. However, it should be long enough to minimize at least surface climate trends.**

**The modelling groups are responsible for a comprehensive documentation of the model system and the experiments. Documentation should be provided via the ESDOC website and tools provided by CMIP6 (http://es-doc.org/) to facilitate communication with other CMIP6 projects. A PMIP4 special issue in GMD and Climate of the Past has been opened where the groups are encouraged to publish these documentations.**

**The documentation should include:**
- **The model version and specifications, like interactive vegetation or interactive aerosol modules etc.**
- **A link to the DECK experiments performed with this model version.**
- **Specification of the forcing data sets used and their implementation in the model. The provision of figures and tables giving monthly-latitude insolation anomalies and daily incoming solar radiation at the top of the atmosphere (TOA) for one year should be provided because this allows the implementation of the most critical forcing to be checked.**
- **Information about the initial conditions and spin-up technique used.**
- **We request providing information on drift in key variables for a few hundred years at the end of the spin-up and the beginning of the actual experiment. These variables are:**
  **- globally and annually averaged SSTs**
  **- deep ocean temperatures (global and annual average over depths below 2500m)**
  **- deep ocean salinity (global and annual average over depths below 2500m)**
  **- top of atmosphere energy budget (global and annual average)**

**- surface energy budget (global and annual average)**
**- northern sea-ice (annual average over northern hemisphere)**
**- southern sea-ice (annual average over southern hemisphere)**
**- northern surface air temperature (annual average over northern hemisphere)**
**- southern surface air temperature (annual average over southern hemisphere)**
**- Atlantic Meridional Overturning Circulation (maximum overturning stream function in the North Atlantic basin between 0 and 80°N below 500m depth)**
**- carbon budget (if relevant).**

8. *We are all confused about what the comment that groups are responsible for finding their own ESGF space for Tier2-3 experiments means in practice, and note that there is no indication as to whether the LGM sensitivity experiments should be uploaded or not.*

   *\* Please clarify what, in practice, this group responsibility means. Will modellers be able to upload Tier 2 and 3 experiments to ESGF or not, and how are the LGM sensitivity runs to be made available?*

On the PMIP side, we are taking all necessary measures so that the PMIP Tier-2 and Tier-3 output can be uploaded and published on the ESGF distributed network. However, LSCE, who is coordinating the database, cannot provide the disk space for all modelling groups as it did until PMIP2, and for some of the groups in PMIP3. This is what we meant by the statements in the papers. Modelling groups participating to PMIP will have to coordinate with their national ESGF node to upload their data on ESGF, as they will do for their PMIP4-CMIP6 Tier-1 data. We have clarified this in Section 6 of our paper.

**Data from PMIP4-only Tier 2 and 3 simulations must be processed following the same standards as Tier 1 for data processing (e.g. CMOR standards) and should be distributed via the PMIP4 ESGF or the CMIP6 ESGF Tier 2 and Tier 3. Modeling groups producing these simulations are responsible to secure suitable space on ESGF nodes. These experiments will follow the same naming, variable convention and format, and documentation requests as Tier 1 PMIP4-CMIP6 experiment so as to be compliant with ESGF database requirements.**

9. *Are all boundary conditions to be uploaded to ESGF? This is promised in the LGM paper, but not in the others.*

   *\* Please make it clear whether all boundary conditions will be uploaded to ESGF. It would be best if they were, as the current situation where everything is available on the PMIP website is sub-optimal (it seems to me OK for modellers doing the runs now, but it is not future-proof: web addresses change!).*

We have put a link in the ESGF/Input4MIPs reference document providing the connection to the PMIP4 web site. This way it is also done for other CMIP6 experiments (see the reference document entries for FAFMIP, OMIP, VolMIP etc.). However, owing to our priority to make the data sets available as fast as possible, we didn't have resources to make the data sets fully

Input4MIPs compliant. We will work with the providers of the data sets towards a distribution via ESGF.

We have included this information in the text:

**The forcing and boundary condition data sets described in this paper are available in the PMIP4 repository https://pmip4.lsce.ipsl.fr/doku.php/exp_design:index. After final acceptance of this manuscript, they will be made available also through Input4MIPs (https://esgf-node.llnl.gov/projects/input4mips/, see the living document "Input4MIPs summary" there on the progress of this process).**

*10. The LGM experiment requires 100 years of equilibrated run to be stored on ESGF, and defines which variables are to be used by participating groups to ensure sufficient equilibrium is attained before the other experiments are started. There is less detail on this in the other papers.*

*\* Please can all experiments require the 100 years of equilibrated run on ESGF. If possible, please define a common metric for assessing equilibrium for all the runs. If there are good reasons why different metrics should be used for the different experiments, please provide an explanation and guidance for the modellers.*

For the Interglacial experiments, we have defined the spinup and metrics for assessing equilibrium of the experiments. See new text for Section2.6 included in response #7 above.

We also now better define what data should be uploaded to the ESGF in revised Section 6

[revised manuscript text omitted]

---

## Author Response (AR3)

**Additional Response to Topical Editor's comments on "The PMIP4 contribution to CMIP6 - Part 2: Two Interglacials, Scientific Objective and Experimental Design for Holocene and Last Interglacial Simulations" by Otto-Bliesner et al.**

Topical Editor comment:

1. *The second response to reviewer #3 does not seem to reflect on the question, is that a copy/paste mistake?*

Reviewer #3 comment

a) The authors correctly highlight uncertainties arising from prescribing or simulating Holocene and Eemian vegetation patterns. The authors recommend using the reconstruction by Hoelzmann et al. (1998) for Holocene North Africa. Is this still the best reconstruction? What about the reconstructions mentioned in the papers cited by the authors or by Lézine et al. (2011), Larrasoana et al. (2013) , : : :? Perhaps there are good reasons to still use Hoelzmann's et al data. But this should be critically reassessed.

*A critical assessment of the various vegetation reconstructions for Holocene North Africa is beyond the scope of this experimental design paper in light of the reduced Tier 2 PMIP4 simulations in the revised paper. We agree that there are other vegetation reconstructions than that of by Hoelzmann et al. (1998) for Holocene North Africa as noted in our initial submission of this manuscript as well as those mentioned by Lézine et al. (2011), Larrasoana et al. (2013), and others. Even if Hoelzmann might not be the best reconstruction, there is still a lack of a reconstruction to use in these regions as boundary conditions for climate models. We proposed this one, because previous simulations are available using this reconstruction, so that model results could also be compared with this previous results. However, as recognised by the reviewers there were too many sensitivity experiments proposed as part of this working group, which explains why we decided to only propose idealised experiments. These idealised experiment are there to understand model results, not to propose realistic simulations. We hope that during the course of the PMIP4 phases other reconstructions will be made available and tested by a subset of modeling groups directly working on this subject.*